



# A monitoring system for spatiotemporal electrical self-potential measurements in cryospheric environments

Maximilian Weigand[1], Florian M. Wagner[2], Jonas K. Limbrock[1], Christin Hilbich[3], Christian Hauck[3], and Andreas Kemna[1]

[1]Geophysics Section, Institute of Geosciences, University of Bonn, Bonn, Germany
[2]Institute for Applied Geophysics and Geothermal Energy, RWTH Aachen University, Aachen, Germany
[3]Department of Geosciences, University of Fribourg, Fribourg, Switzerland

**Correspondence:** Maximilian Weigand (mweigand@geo.uni-bonn.de)

**Abstract.** Climate-induced warming increasingly leads to degradation of high-alpine permafrost. In order to develop early warning systems for imminent slope destabilization, knowledge about hydrological flow processes in the subsurface is urgently needed. Due to the fast dynamics associated with slope failures, non- or minimally invasive methods are required for cheap and timely characterization and monitoring of potential failure sites to allow in-time responses. These requirements can potentially
be met by geophysical methods usually applied in near-surface geophysical settings, such as electrical resistivity tomography (ERT), ground penetrating radar (GPR), various seismic methods, and self-potential (SP) measurements. While ERT and GPR have their primary uses in detecting lithological subsurface structure and liquid water/ice content variations, SP measurements are sensitive to active water flow in the subsurface. Combined, these methods provide huge potential to monitor the dynamic hydrological evolution of permafrost systems. However, while conceptually simple, the technical application of the SP method
in high-alpine mountain regions is challenging, especially if spatially resolved information is required. We here report on the design, construction, and testing phase of a multi-electrode SP measurement system aimed at characterizing surface runoff and melt-water flow at the Schilthorn, Bernese Alps, Switzerland. Design requirements for a year-round measurement system are discussed, the hardware and software of the constructed system, as well as test measurements are presented, including detailed quality assessment studies. On-site noise measurements and one laboratory experiment on freezing and thawing characteristics
of the SP electrodes provide supporting information. It was found that a detailed quality assessment of the measured data is important for such challenging field site operations, requiring adapted measurement schemes to allow for the extraction of robust data in light of an environment highly contaminated by anthropogenic and natural noise components. Finally, possible short- and long-term improvements to the system are discussed and recommendations for future installations are developed.





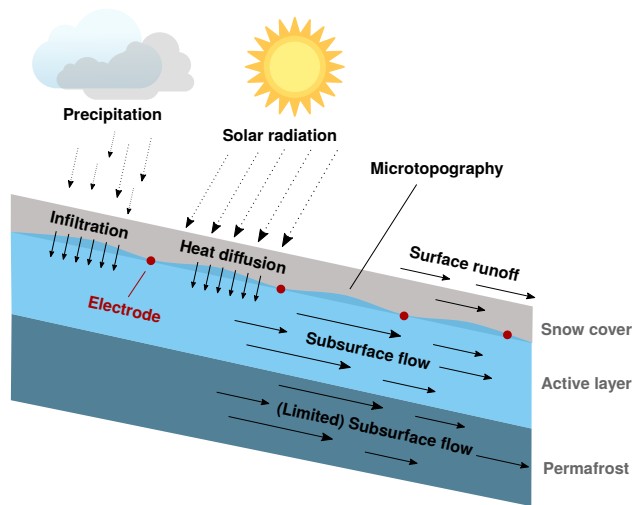

**Figure 1.** Sketch of natural and anthropogenic SP signatures expected for measurements in permafrost environments.

# 1 Introduction

Long-term borehole temperature records document the degradation of permafrost on a global scale (Biskaborn et al., 2019), which can release soil organic carbon into the atmosphere (e.g., Schuur et al., 2015) and negatively affect slope stability in alpine regions (e.g., Huggel et al., 2012). The state and evolution of high-mountain permafrost regions is therefore of interest

for climate and cryospheric research, as well as for risk-management in these regions. As such it is increasingly important to provide cheap and fast characterization and monitoring facilities for regions important to human settlements, infrastructure, and the environment. Traditionally, point measurements of relevant data, such as temperature, ice content, and snow cover thickness, provide the basis for research, whereas remote sensing provides data on surface displacement and actual mass movement events. Non-invasive geophysical imaging techniques, such as electrical resistivity tomography (ERT), ground penetrating

radar (GPR), and seismic refraction tomography have been increasingly investigated to provide structural information on the subsurface of permafrost regions (e.g., Maurer and Hauck, 2007; Hilbich et al., 2008; Hubbard et al., 2013; Merz et al., 2015; Wagner et al., 2019). However, all these methods have in common that they are sensitive to structure only, although limited indirect information on processes can be inferred by time-lapse monitoring setups (e.g., Hilbich et al., 2008, 2011; Oldenborger and LeBlanc, 2018; Mollaret et al., 2019). In contrast, the self-potential (SP) method offers the unique possibility to observe

active water flow by means of generated electrical streaming potentials (e.g., Telford et al., 1990; Revil et al., 2012), and is listed among the geophysical methods of high potential interest to permafrost research (e.g., Hauck, 2013).

Various sources of naturally occurring electrical potentials have been identified: electrokinetic (streaming) potentials, electrochemical potentials, thermoelectric potentials, and telluric potentials (e.g., Telford et al., 1990; Revil et al., 2012). Figure 1 visualizes the various elemental drivers and their corresponding impact on resulting SP signals, as expected for alpine environ-

ments. Specifics on sources of SP can be found in a wide range of literature, for example Blake and Clarke (1999) provide a





brief, but comprehensive, description of streaming potentials. For the sake of brevity we here provide only qualitative descriptions of the streaming, diffusion, and magneto-telluric (MT) potentials, meant only as a primer to understand the basic premise of our undertaking.

Fluid flow in porous media generates electrical potential gradients, so-called streaming potentials, commonly anti-parallel
to the fluid pressure gradient (e.g., Telford et al., 1990; Bernabé, 1998; Revil et al., 2012). The electrical charge separation, required to build up these potentials, happens at the shear plane of the so-called electrical double layer (EDL), which is a multi-layered ion concentration gradient structure that forms at the interface of charged surfaces and bipolar liquids (e.g., Revil et al., 2012). Due to the non-uniform charge distribution in the pore space, fluid flow leads to an effective charge separation in flow direction. Macroscopically, the relation between pressure difference $\Delta P$ and ensuing electrical potential difference $V$
is described by the Helmholtz-Smoluchowski (HS) equation (e.g., Morgan et al., 1989), $\frac{V}{\Delta P} = c$, with $c = \frac{-\zeta\epsilon}{\mu\sigma_b}$, where $c$ is the streaming potential coupling coefficient, $\zeta$ is the Zeta potential (the electrical potential at the shear plane), $\sigma_b$ is the bulk electrical conductivity of the subsurface (comprised of fluid conductivity and surface conductivity), $\epsilon$ the electrical permittivity of the fluid, and $\mu$ the dynamic viscosity of the fluid. Note that variations in the definition of $c$ exist, and without detailed per-case analysis application of the equation should be limited to qualitative use cases. For example, Blake and Clarke (1999)
introduce empirical factors that account for pore tortuosity and porosity. Also, care has to be taken regarding the sign of $c$, which can potentially flip under certain conditions, such as extreme pH environments (e.g., Leroy and Revil, 2004), leading to potential reversals of apparent flow directions.

The HS equation was initially developed only for capillary flow, but was later shown to be of (limited) validity also in porous media (e.g., Morgan et al., 1989; Bernabé, 1998). In order to formulate the problem for a spatially variable flow, and
thus electrical potential, field, a formulation based on transport equations for representative volumes of electrical and fluid flow is used (e.g., Jardani et al., 2007; Jougnot et al., 2020, and references therein). The inverse problem thus estimates the spatial distribution of hydraulic velocities, based on suitably measured, spatially distributed, SP dipoles (e.g., Jardani et al., 2007; Ahmed et al., 2013).

Diffusion potentials are caused by electrical charge separation in concentration gradients in the pore fluid. Closely related to
this, thermal potentials are caused by temperature gradients, which effectively lead to diffusion potentials caused by differing ion mobilities. Large-scale electrical potentials can also be caused by magneto-telluric currents in then ionosphere, which induce electrical currents into the subsurface, and manifest in periodic atmospheric and subsurface processes (e.g., Egbert and Booker, 1992; MacAllister et al., 2016).

Passive SP signatures have been used to assess a multitude of different research questions, such as characterizing pumping
experiments (e.g., DesRoches and Butler, 2016; DesRoches et al., 2018, and references therein), assessing earthquake triggers (e.g., Bernabé, 1998; Guzmán-Vargas et al., 2009, and references therein), characterizing dams (e.g., Moore et al., 2011), and the monitoring of volcanoes (e.g., Maio et al., 1997; Friedel et al., 2004). Water flow in and around trees has also been investigated using the SP method (e.g., Gibert et al., 2006; Voytek et al., 2019). There is also a steady interest in characterizing landslides using active and passive electrical methods (e.g., Bogoslovsky and Ogilvy, 1977; Naudet et al., 2008; Heinze et al.,
2019; Whiteley et al., 2019, and references therein).





Knowledge about hydrological processes in the subsurface, and potentially below a solid snow cover, would be greatly beneficial to thermo-hydraulic modeling and stability assessment of sloping permafrost regions. Scapozza et al. (2008) presented SP measurements on rock glaciers, finding evidence that SP profiles can provide information on water content and movement. Thompson et al. (2016) investigated melt water flow in snow packs over a glacier using SP, finding clear evidence for streaming

potentials generated by melt water within the snow packs. Voytek et al. (2016) presented combined ERT and SP measurements on a arctic hill-slope, finding that SP potentials follow topography and expected flow paths on larger scales, while showing much more complex behavior on smaller scales.

Relatively few applications of long-term SP monitoring setups for mountainous regions have been reported: Blake and Clarke (1999) report on multi-year SP measurements at the bottom of a glacier in Canada, finding clear evidence for electrical

streaming potentials caused by periodic and episodic water movement at the bed of the glacier. Trique et al. (2002), and associated publications, report on a multi-year SP measurement campaign in the French Alps. Electrical signature changes were associated with changes in the water level of adjacent reservoir lakes, which in turn were related to vertical water movement in the soil. Kulessa et al. (2003) present results from multi-year borehole SP measurements in a glacier, successfully relating electrical signals to water flow direction and velocity, and investigating the separation of geochemically and hydraulically

generated electrical signals. Other notable long-term monitoring studies are often related to volcanic research, in which natural electrical signals are investigated as precursor signals to seismic activity. For example, Friedel et al. (2004) present SP results from the Merapi volcano, finding evidence for periodic and episodic water movement caused by atmospheric changes and speculate about transients induced by volcanic activity. Recently, Hu et al. (2020) discussed a four-year multi-electrode time series of borehole SP measurements in a hydrological observatory, comparing measured data with modeled SP data, while

taking into account water flow and transport processes.

Despite these promising studies, SP measurements in mountainous and permafrost regions remain relatively scarce (e.g., Hauck, 2013), possibly due to the problems and ambiguities arising in the analysis of SP data with regard to subsurface flow parameters in these complex environments. Correspondingly, equipment for SP measurements in these regions is not readily available, requiring improvisation and self-assembly, hindering progress in the assessment of the SP method for permafrost

research.

While the SP method is still in regular use, we observe that studies dealing with technical details of the method are often quite dated, with the associated danger of information being lost to time and the general flood of scientific literature, as well as being outdated due to modern technological advances. For example, Corwin (1989) discuss various aspects of mobile SP procedures, as well as monitoring aspects, while basing some of their suggestions on technical details of the epoche. Perrier

et al. (1997) present valuable information on electrode characteristics, long-term stability, and suggested installation methods, as well as a discussion of useful long-term monitoring preparations. With respect to technical aspects of long-term electrode behavior in unsaturated environments, Hu et al. (2020) offer important insight into the electro-chemical details that lead to electrode effects superimposing on the process signals. They conclude that explicit electrode modeling is required in order to explain long-term monitoring SP signatures. A more stationary, long-term monitoring of natural electrical signals was reported

and technically discussed by Blum et al. (2017).





In this study we report on our experiences in planning, designing, building, and deploying an SP measuring system at the Schilthorn, Bernese Alps, Switzerland. Given the importance of reliable permafrost monitoring, the potential of the SP method, and the lack of commercially available systems for spatio-temporal monitoring of SP and temperature, this study documents useful findings for the development and deployment of SP systems particular for, but not limited to, the emerging cryospheric geophysical community.

We start by formulating the scientific aims for SP measurements and the associated system requirements that guided the technical design of the measurement system, presented in the section thereafter. Then, quality assessment strategies, as well as data from a multi-month test run over the winter season 2017/2018 are presented and discussed. We do not attempt to go into detail regarding the process information contained in the data, leaving this analysis for future, separate studies. We finish with a short discussion on possible improvements and alternative approaches for future system designs.

## 2 Scientific aims and resulting system requirements

The primary function of the proposed measurement system is the long-term monitoring of electrokinetic streaming potentials in alpine permafrost regions. Electrical potentials are measured with respect to a given reference potential in the form of electrical voltages. These dipoles shall be related to the spatial distribution (flow patterns) of near-surface fluid flow, triggered by either precipitation, snow melt, or freezing/thawing events (see Fig. 1), to be ultimately of help for structural and cryo-hydrological modeling frameworks. In the context of climate warming, subsurface (permafrost) thawing is of special interest.

The aforementioned goal of characterizing subsurface fluid flow directly translates into multiple requirements for measurement equipment: A long-term, year round, unsupervised operation must be ensured to minimize human intervention, which is not easily and timely done in these regions and at all times of the year. Correspondingly, the system should be easily maintainable on-site in the summer months, and should be as modular as possible to reduce the amount of material to be transported. Also, the system must operate autonomously on battery power, charged by solar or fuel cells, and should be able to adapt a measurement scheme according to changing power constraints. All-year weather, with large temperature variations (and corresponding changes in humidity), freezing, extreme precipitation, lightning, snow, and strong winds, require a corresponding protection against the environment. Expected levels of snow cover should not structurally affect the measurement system.

Measurement data should be wirelessly uploaded from the device to ensure timely quality assessment of the data, integration into near-realtime analysis frameworks, and to plan maintenance in advance based on ongoing monitoring of system health. Optimally, two-way communication would allow reprogramming of the logging system remotely, reducing further costly on-site visits.

Given the small scientific community currently working on the measurement of SP data in permafrost regions, system components should be mostly "off the shelve" and minimize high-tech, custom-built electronics as far as possible.

SP measurements exhibit inherent ambiguities with regard to relevant hydraulic parameters. Therefore, additional parameters must be collected by the measurement system, or by coordinated other means: To properly take into account temperature effects on the electrodes and in the soil, temperature data must be available, at least at the location of each electrode. Electrical





conductivity distributions can be recovered using time-lapse ERT applications, but care should be taken to cover the relevant subsurface volume. Past studies have also greatly benefited from magnetic measurements for assessment of magneto-telluric signal components, water pressure sensors (e.g., Blake and Clarke, 1999), and pH probes. Given a suitable understanding of the geochemical subsurface composition and evolution, laboratory experiments on selected samples could be used to estimate
coupling coefficients, used to convert the SP signatures into flow information.

Finally, weather information such as snow cover height and precipitation data can be very helpful in assessing the overall state of the physical system under investigation, and to relate (extreme) weather events to corresponding (transient) electrical signals.

## 3   Site description and history of on-site geophysical measurements

The SP monitoring system was installed at the Schilthorn field site, located in the Bernese Oberland in the northern Swiss Alps. The study area is located at the north-facing slope, which has an inclination of $30°$, approximately 60 m below the ridge at about 2900 m asl. The lithology in the Schilthorn massif mainly consists of dark, micaceous shales with interbedded sandstones (Glockhaus formation), interbedded fine-grained clastic marly limestones and marls (Schwarzhorn beds) (Imhof et al., 2000). Around the test site, the surface consists of small to medium size debris including a layer of up to 5 m comprised of sandy, silty
and clay material originating from the weathered bedrock, here micaceous shales (Hauck, 2002; Hilbich et al., 2008; Scherler et al., 2013; Pellet and Hauck, 2017). The main morphology of this permafrost site is described as crest. There is no more glaciation at the Schilthorn, with long-lasting snow patches only occurring in some years. Most of precipitation falls as snow (Hilbich et al., 2008; Hoelzle and Gruber, 2008), with up to 3 m snow thickness and recurrent avalanche risk. Typically, the snow melt period at Schilthorn north facing slope starts in early May (PERMOS, 2019). The permafrost is >100 m thick, and
mainly isotherm with very little ice content. The ongoing permafrost degradation is reflected by the increase of the maximum active layer thickness from ca. 5 m at the beginning of borehole measurements to up to 10 m depth in 2018 (PERMOS, 2019). Detailed descriptions of the geology and the permafrost distribution around the Schilthorn summit and detailed meteorological and geophysical monitoring data can be found in Imhof et al. (2000); Hilbich et al. (2011); Scherler et al. (2010, 2013); Pellet and Hauck (2017) and PERMOS (2019).
As part of the PERMOS network, ground surface temperature measurements have been performed since 1999 at the Schilthorn in three boreholes. In addition, soil moisture measurements, as well as refraction seismic tomography and ERT measurements have been conducted in regular intervals (Pellet and Hauck, 2017; PERMOS, 2019). Meteorological data, such as air and surface temperature, relative humidity, solar radiation, snow height, as well as wind speed and direction, are recorded at the meteorological station next to the borehole location (Hilbich et al., 2008, 2011; Hoelzle and Gruber, 2008). A horizontal
survey line in east-west direction for ERT monitoring was established in 1999, automatized in 2009 and extended in 2012 (Hauck, 2002; Hilbich et al., 2011; Mollaret et al., 2019).


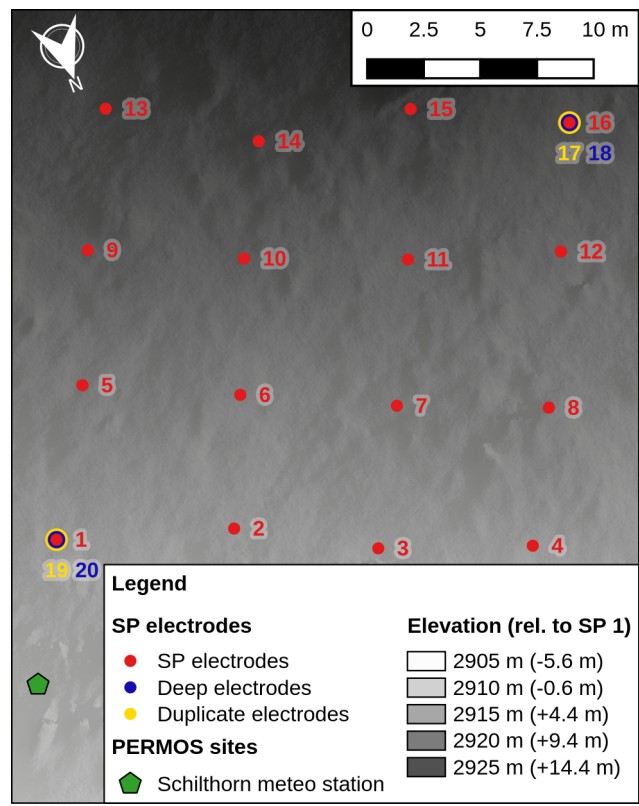

**Figure 2.** Electrode layout at the Schilthorn summit. Electrodes are roughly 7.5 m apart in horizontal and vertical directions. Deep electrodes 18 and 20 are located 20-30 cm below the reference electrodes 1 and 16, respectively, while electrodes 17 and 19 duplicate the reference electrodes 1 and 16, respectively, in only a few cm distance (see Fig. 3 for photograph of electrodes 1 and 19). The legend denotes absolute electrodes heights, as well as heights relative to electrode one.

A cable car station is located at the Schilthorn summit, in a distance of about 100 m to the field site, including tourist infrastructure in the form of a restaurant, viewing platforms, footpaths along the ridge, and off-piste skiers sometimes directly passing the field site during winter.

# 4 System design and installation

5 In this section the design of the SP system is presented. This includes the installation of electrodes at the test site, and associated measurement considerations, as well as the actual construction of the measurement system.





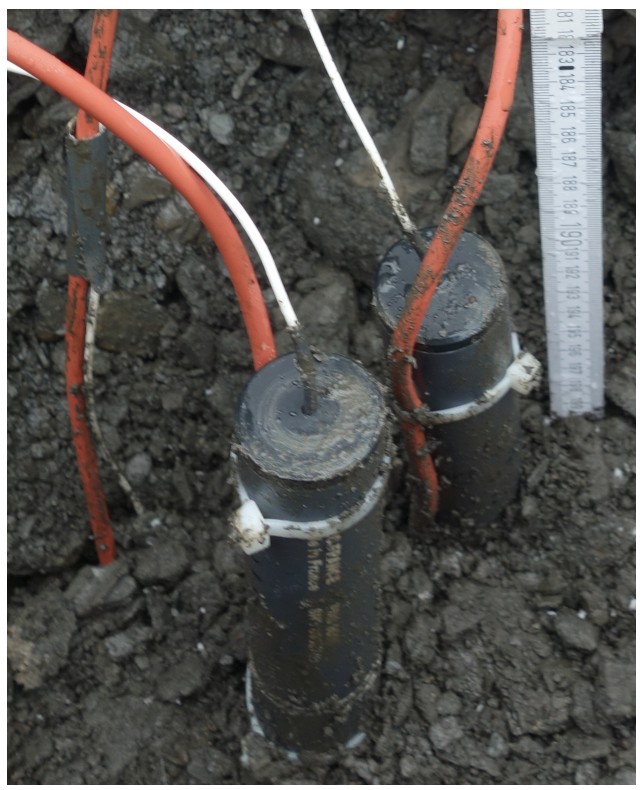

**Figure 3.** Installation of electrodes 1 and 19. Electrode 20 is located directly below these both electrodes, indicated by the feeding cables on the left. White cables are connected to the electrodes, while red cables house four stranded wires leading to Pt100-temperature sensors at the tip of each electrode.

## 4.1 Electrode installation and measurement schedule

The SP setup consists of 20 unpolarizable *Pb/PbCl$_2$* SP-electrodes of the Petiau-type (Petiau, 2000), arranged in a 4 x 4 grid of 16 electrodes with an approximate lateral and vertical distance of 7.5 m (Fig. 2). Another two electrodes are located near the lower left electrode (no. 1), with one electrode (no. 19) used to replicate electrode no. 1, and the other electrode (no. 20) located approximately 20-30 cm deeper than electrode 1 in the soil. Electrodes 1 and 19 are shown in figure 3. A similar arrangement is implemented for electrode no. 16, with electrode no. 17 duplicating it, and electrode no. 18 buried below both of them. Electrodes were placed at 30-40 cm below the surface to reduce diurnal temperature effects caused by direct solar radiation, and to ensure proper contact with the soil. We refrained from embedding the electrodes in bentonite clay, due to unknown freezing/thawing behavior, and because salt leaching and corresponding electrical diffusion/concentration potentials can be expected in saturated flow regimes (e.g., Perrier et al., 1997), as expected for the test site. As discussed by Hu et al. (2020), even in non-freezing, unsaturated, conditions it seems that the used electrode type will degrade relatively fast. Coupled to the fact that it is still unclear how the unpolarizable electrodes will age in continuous freezing/thawing environments, it is


important to carefully monitor electrode performance. In the preceding years of experiments with SP electrodes in permafrost regions, we sometimes encountered loosened membranes at the tips of the electrodes. However, these electrodes were usually not properly embedded into the soil, thereby missing corresponding counter-forces under thermal changes, or experienced some kind of physical force due to improper installation. Proper embedding into the soil ensures proper counter-forces during

freezing, which should prevent the membrane from being pushed out by frozen electrode slush. As recommended, e.g., by Corwin (1989), we performed base-line offset measurements of the SP electrodes in saline solution. Out of a pool of available SP electrodes, electrodes were selected ('matched') to minimize their baseline offset when measured within a saline solution. A random selection of SP electrodes initially yielded baseline measurements of up to -5 mV (with respect to the reference electrode), while matching reduced this effect to within ±1mV (also visualized in Fig. S6). Each electrode is fitted with a

Pt100 temperature sensor, located directly at the electrode membrane (see also pictures in Fig. 6a).

Measurements of electrical SPs were conducted with respect to the reference electrode no. 1 (lower left corner), resulting in 15 relative potentials (voltages). To increase system reliability, and provide means for consistency checking, SP potentials were also measured with respect to electrode 16, which is located at the upper right corner of the electrode grid. The potentials between reference electrodes and their duplicates (electrode pairs 1-19 and 16-17), located only a few centimeters apart,

were also recorded to analyze electrode drift and localized inhomogeneities, as were potential differences between reference electrodes and their associated electrodes at deeper locations (pairs 1-20 and 16-18).

Electrical voltages and various operational data (battery voltage, solar voltage, internal logger temperatures) were measured every 10 minutes at daytime, and every 30 minutes at night time, in order to conserve power. To reduce aliasing effects, voltage measurements were repeated, and averaged, 10 times, with the internal integration time of the data logger set to 50 ms (20 Hz).

Once per hour, temperatures and voltages across the temperature sensors were measured. In order to prolong measurement operations for low-power situations, depending on the battery voltage, these measurement intervals were reduced to 1 hour measurements for a battery voltage between 12.3 and 12.5 volts, and to 6 hours for battery voltages below 12.2 volts. Once per day the resistances between horizontally adjacent electrodes, as well as vertical pairs at the beginning of each electrode row (i.e., pairs 1-5, 5-9, 9-13, see Fig. 2) were recorded using a 2.5 $\mu$A constant current source for excitation. This measurement

is referred to as the contact resistance between the two involved electrodes, and incorporates the interface resistance between electrodes and soil, as well as the soil resistance itself.

System time was set to UTC in order to prevent mismatches in time stamps with external data sources, and measurement data was uploaded by email every six hours. Once per day the system updated its internal real-time clock using the network time protocol (NTP) in a scheduled online session. The detailed measurement schedule can be found in table S1, and the

programming for the data logger can be found in listing S1.

## 4.2 Composition of the measurement system

The overall system design, as constructed and deployed in the years 2017-2019, is presented in Fig. 4. Power is provided by two 12V Exide ES950 gel batteries, connected in parallel to reach a design capacity of 190 Ah. A 30-watt solar panel is used to charge the batteries using a Steca Solsum solar charger. Data is logged by a Datataker DT80M, Series-4 data logger, to which





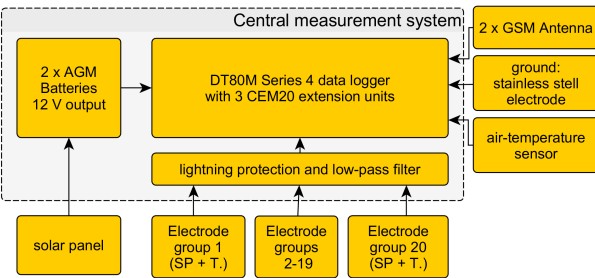

**Figure 4.** Schematic overview of essential system components. Components included in the group "Central measurement system" are housed in an aluminium box (see also Fig. 5), while all other components are external and connected by cables. The term 'electrode group' refers to each pair of SP-electrode and attached temperature sensor. The components termed 'lightning protection and low-pass filter' were only installed and tested in 2019.

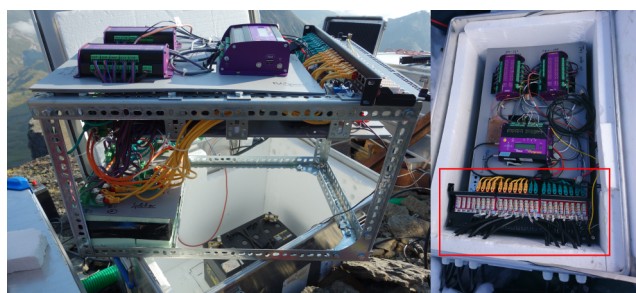

**Figure 5.** Central measurement system. The red rectangle marks the patch panel, at which the incoming connections from the electrodes are transferred to standard ethernet cables.

three Datataker CEM20 extension modules are connected, resulting in 62 available analog-in channels. Each input channel of the data logger consists of 4 terminals, allowing, among others, one four-point resistance measurement, or up to 4 voltage measurements between different terminal pairs. It was found that the system operates more reliably under cold conditions if power to the logger was increased from 12V to 24V using a DC-DC converter. A metal rod inserted into the soil a few meters away from the system was used as system ground.

Two GSM antennas are connected to the data logger, providing improved signal reception capabilities. Apart from one Pt100 temperature sensor used to measure air/snow temperature, directly attached to the box housing the measurement system, each unpolarizable SP electrode is paired to one Pt100 temperature sensor, with both sensors referred to as an 'electrode group'. Each electrode group is connected and grounded to the measurement system via a 8-wired shielded ethernet cable resistant to UV radiation. The actual measurement system is housed in an aluminium case that can withstand expected snow covers, and is lined with Styropor on the inside for thermal shielding (Fig. 5).

The ethernet cables of each sensor package are introduced into the box via water-tight IP-cable glands and connected to an ethernet panel via standard insulation displacement connectors (IDC). Signal lines from 20 electrode groups are internally





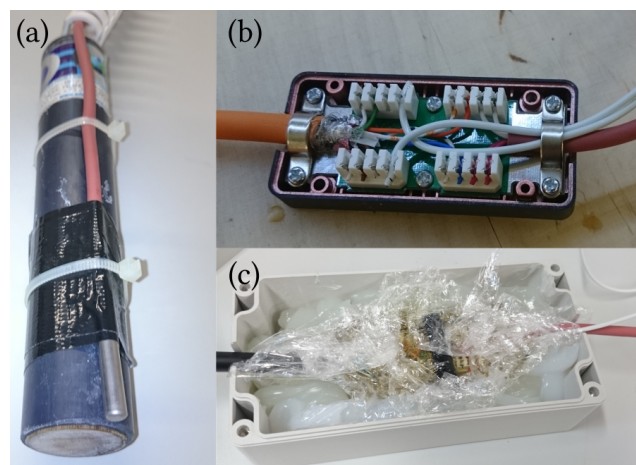

**Figure 6.** Installation of SP electrodes a) $PbCl_2$ electrode with attached Pt-100 temperature sensor. Note that this specific electrode depicted in the picture was only used for construction testing, and was not used for SP measurements due to the dried-out membrane. b) Connection box in which the electrode/temperature cables are connected to the ethernet cable used for long-distance signal transport, c) connection box from b), embedded in silicon within a sturdy PVC box for physical protection.

routed and multiplexed to the inputs of the logging system. In the summer of 2019 a lightning protection circuit based on ESD protection diodes was tested, protecting all SP input channels against voltage surges caused by nearby storms.

As discussed by DesRoches and Butler (2016), the internal filtering capabilities of data loggers are not always sufficient to suppress high-frequency noise superimposed on the, usually weak, SP signals, which prompted them to install discrete low-
pass filters based on R-C-circuits at the input channels of the data logger. DesRoches and Butler (2016) also pointed out that the cut-off frequency of simple R-C-filters depends on the soil resistance. Our 2019 tests also included a low-pass filter (cut-off frequency 1 Hz) to reduce the amount of aliasing noise. Due to technical restrictions, the cut-off frequency of the low-pass filter could not be adapted to the Nyquist limit of the measurement schedule. As such, some remaining data noise caused by aliasing of high-frequency components is to be expected. Routing and multiplexing within the SP-system is implemented using
custom-made PCBs, interconnected using ethernet cables. Figure S1 depicts the signal flow within the system, and Figs. S3, S4, S5 show the wiring logic of the three, interconnected, PCB boards, ending in the individual data logger input channels.

The electrode groups are connected to the ethernet cables using small switching boxes, again using IDC technology, located a few centimeters next to each sensor (Fig. 6b). To prevent any contact issues caused by humidity, the connection box is sealed with two-component epoxy resin (rated for use below $0°$). A planned, but only partially implemented, next step is to embed
this resin-covered box in a plastic case filled with silicon to provide additional physical protection (Fig. 6c).





## 5   First measurement results

This section introduces preparatory and auxiliary measurements, as well as first data results for SP and system-health information. We do not attempt a comprehensive scientific analysis of the data with respect to hydrological flow parameters, as this would go beyond the scope of the manuscript. As such, results are presented and discussed with the aim of assessing the system
operations, quality, and the suitability for long-term scientific applications.

Presented SP data was filtered using a rolling mean filter to suppress high-frequency noise components. While not in the direct scope of this study, Sect. S4 in the supplement shortly discusses the applied filters and potential alternatives to those used here.

### 5.1   Auxiliary measurements

In order to better assess the system operations and expected measurement results, two additional experiments were conducted: One laboratory experiment was designed to investigate electrical potentials generated within electrodes during freezing and thawing under no-flow conditions. The other experiment was a high-frequency noise measurement, conducted on site at the Schilthorn in the summer 2019 using the largest available SP dipole, aimed at investigating the magnitude and stability of noise components encountered during routine operations of the monitoring system.

### 5.1.1   Laboratory investigation: electrode effects during temperature cycling

Not much is known about the short- and long-term characteristics of unpolarizable SP electrodes under repeated freeze/thaw cycles in permafrost environments. While not exaustive, and meant more as a first qualitative advance, a simple laboratory experiment was conceived and executed to investigate the electrical behavior of SP electrodes in controlled freeze/thaw cycles (Fig. 7a): Two electrodes were fully embedded into the soil, while another two were only partially embedded. Regardless of
the electrode position, all membranes (at which external potentials are sensed) were located in the soil, close to each other. The setup was then exposed to a freezing and a thawing cycle in a climate chamber, while the electrical voltages between electrodes were measured.

Significant voltage anomalies with different signs were observed during freezing and thawing, with voltages between one embedded and one standing electrode considerably higher than between both embedded and both standing electrodes (Fig.
7c, potential between both standing electrodes not shown). We explain this observation with concentration/diffusion potentials created within the standing electrodes during temperature changes, caused by different thermal characteristics of electrode and soil. This is further supported by the observed surface temperature difference between electrodes and soil via thermal camera (Fig. 7b). An additional component, expected in all measurements, are freezing potentials, caused by the onset of freezing in the sample (e.g., Rao and Hanley, 1980; Hanley and Rao, 1980). In addition, even in climate chambers it can be expected
that the soil sample will freeze and thaw non-uniformly, which could lead to changes in pore pressure, fluid redistribution, and corresponding changes in fluid chemistry. All these processes can in turn generate electrical potentials, which we expect





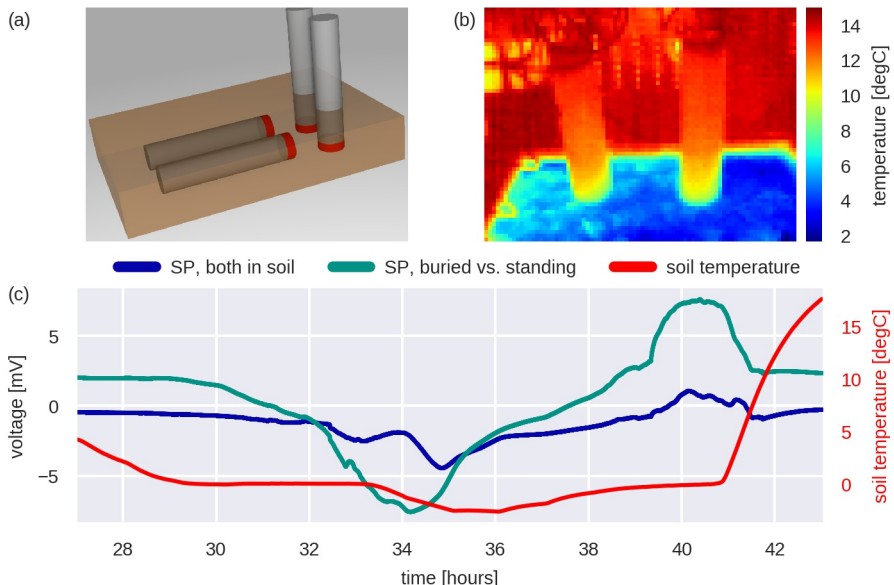

**Figure 7.** Laboratory experiment investigating electrode effects during freezing and thawing. a) Sketch of experimental setup. The container is 30 cm in length, 20 cm in depth, and 5 cm high. Electrodes are shown in gray, with the wood membranes in contact with the surrounding highlighted in red. b) Thermal image (surface temperatures) during thawing c) Electrical voltages and soil temperatures; vertical black line indicates time of thermal image in b)

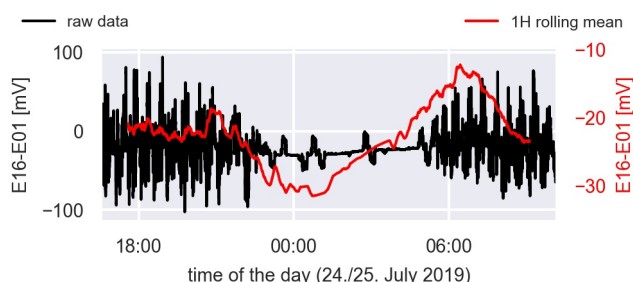

**Figure 8.** Noise measurement on dipole E16-E01, conducted in the summer of 2019. Raw data is drawn in black, a 2 hour moving average filtered version in red.

to contribute to the observed transient SP signals and should be kept in mind when analyzing transient signals formed during freezing and thawing situations with respect to water flow.





### 5.1.2 SP noise measurements

In order to investigate the noise environment in which the site operates, in summer 2019 a 17.5 hour noise measurement with 1 s measurement intervals was conducted on the largest dipole of the system, between electrodes 1 and 16 (for electrode positions, refer to Fig. 2). A low-pass filter with cut-off frequency 1 s was installed. As such, we expect no significant aliasing in this

noise measurement. The time series shows three distinct phases (Fig. 8): The first phase, ending approximately at 22:00 hours, shows large noise components of 30 and 60 minutes. After that, during night-time hours, a relatively stable signal is measured, with three intermittent signal spikes. Shortly before 6:00 the next day, noise levels return to those of the previous day.

We interpret the large noise levels at day time with electrical noise generated by the large machinery associated with the operation of the cable car and touristic facilities at the summit. The observed signal levels exhibited are large in comparison

to expected low-frequency SP signals, yet low-pass filtering reveals a clear diurnal signal (Fig. 2, red line). While large noise levels normally would be undesirable, in the context of system tests we see them as an opportunity to make sure the system works even under those conditions. A wavelet-based frequency analysis of the noise data can be found in the supplement section S5.

### 5.2 Quality assessment

In light of the harsh measurement environment, system integrity must be monitored with special care. This is important to plan maintenance and repair activities, but also to flag bad data caused by malfunctions or strong external noise activities (see Fig. 8). Besides the monitoring of regular system activity in the form of data downloads, battery and solar voltages are monitored with respect to expected system behavior (not shown here). In the following we discuss some of the quality assessment procedures implemented for the presented system, targeted at the measurement of SP data.

### 5.2.1 Contact resistances

Daily contact resistances are used as a rough proxy to electrode health, ensuring that an electrical path exists between the two electrodes involved. Successful measurements are dominated by the interface resistance between the electrodes and the soil, with the actual soil resistance being of minor importance (e.g., Corwin, 1989). As such, contact resistance data, and its temporal evolution, can be used to assess the state of the electrode-soil interface. Very low contact resistance measurements could point

towards short-circuits in the system, while failing measurements (outside the measurement range of the data logger), caused by very large resistances, point to missing electrode-soil contact, or broken cabling.

Figure 9 shows the temporal evolution of contact resistances over the whole measurement period of 2017 and 2018. No extreme data points that indicate short-circuits or lost electrode contacts were observed. However, the actual values of the resistances differ with the electrode pairs, with very high resistances (red colors) occurring for some electrode pairs, and

relatively low resistances (blue) showing up intermittently for the electrode pairs 3-4 and 6-7.

It is not clear if, and how, the actual values of the resistances, in contrast to the patterns and extreme values, can be used to assess electrode and soil characteristics. Note that the patters present in Fig. 9 do not necessarily correlate to the tempera-



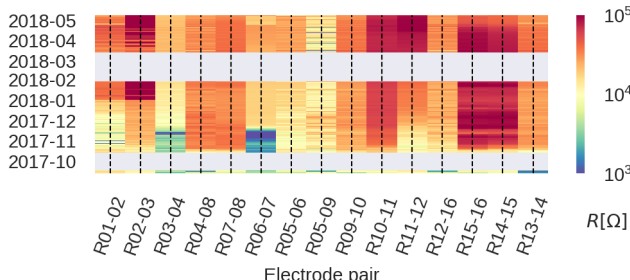

**Figure 9.** Contact resistances (R), measured daily between selected electrode pairs. For electrode numbering, refer to Fig. 2.

tures of the involved electrode temperatures. As such, differences between electrode pair resistances suggest spatially variable subsurface conditions (lithology: grain size, pore size, etc, but also water/ice/air content), while temporal variations indicate changes of pore water chemistry, temperature, and ice content in the direct vicinity of the electrodes.

Note that the use of a static current source leads to electrode polarization, which prevents useful SP measurements with the
involved electrodes for some time after. As such, only one measurement of the contact resistance is conducted per day, and, based on laboratory experience, we suggest to ignore any SP data for up to an hour after these measurements.

Please also note that the measured contact resistances differ from those reported for ERT at the same location (e.g., Mollaret et al., 2019, and references within) for multiple reasons: The construction of SP electrodes is, by design, very different from those used for active methods such as ERT. Active methods require a low internal electrode resistance and a large surface area
to properly inject current into the soil, while passive SP electrodes are only used to measure the electrical potential generated by external means, with the additional requirement of low influence of electrode polarization (e.g., Corwin, 1989; Petiau, 2000). As such, a high internal resistance is, within certain bounds, actually favoured for potential measurements in order to reduce any leakage currents and associated distortion of the measurement. In this regard, the choice of steel electrodes for ERT measurements is a common compromise between good current injection capabilities, and acceptable potential measurements.
Another difference is the surface area: The SP electrodes employed in this study yield only a small membrane of 3 cm diameter in contact with the soil, while the ERT electrodes offer a much larger surface area in soil contact.

### 5.2.2 Superposition of SP dipoles

While contact resistances are easily used to assess electrode contact to the soil, they do not provide information on noise components in the data. Given the large number of signal and noise sources that contribute to the measured SP signatures, ways
to isolate process-related signals from noise components are essential. One way of isolating signatures from noise components is the use of low- or high-pass filtering, given a suitable understanding of the process dynamics. This also implies that the process signal-strength dominates a given frequency band. What follows is the challenge of making sure that noise was suitably removed by the filtering operation.





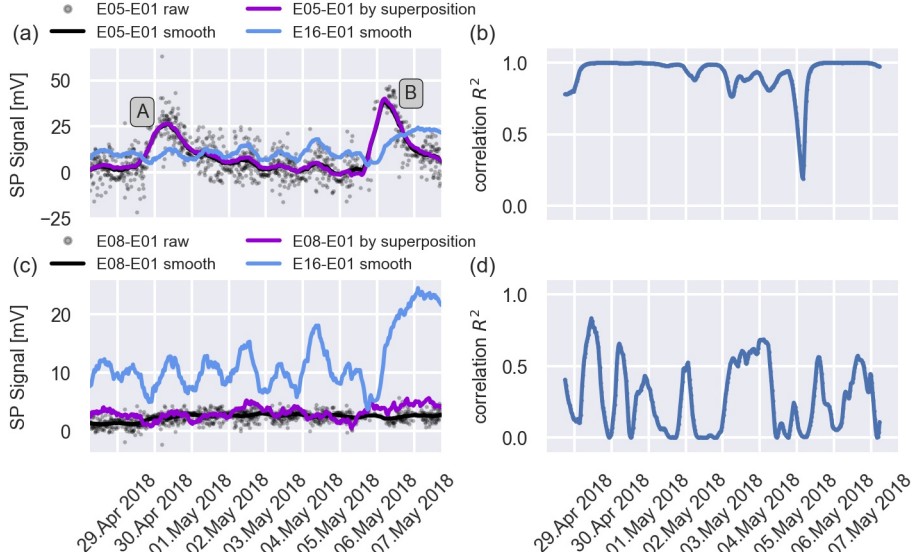

**Figure 10.** Comparison of two SP dipoles, both of which were measured directly and constructed by means of superposition, and corresponding rolling correlation coefficient. Top: Dipole between electrodes 5 and 1, bottom: Dipole between electrodes 8 and 1. a, c) Filtered direct and indirect time-series (moving average filter, window length 12.5 hours). The time-series between both reference electrodes (E16-E01) is shown here for reference, and is used in the superposition procedure. Gray dots indicate raw data of dipoles, from which the rolling mean was computed. Raw data for dipole E16-E01 and for superposition not shown. The labels A and B in a) depict temporally localized anomalies, that will be discussed in detail in Sect. 5.3.3. b, d) Result of moving correlation, window length: 24.5 hours.

The SP potentials are measured redundantly with respect to two reference electrodes, nos. 1 and 16. This provides resilience towards a failure of one of the reference electrodes, but also allows consistency and noise checks by recomputing dipoles via superposition of measurements with respect to the other reference electrode, and subsequent comparison of the results to direct measurements. For example, the dipole E.5-1 can also be computed by superposition: E.5-1$_{super}$ = E.5-16 + E.16-1. Naturally,

noise components will increase in the computed dipole by means of error propagation, leading to different noise levels in direct and indirect measurements. If a signal extraction is now attempted, for example, by applying a low-pass filter to the data, then we can compare the filtered direct and indirect time-series. Any deviation from both time-series can be attributed to incompletely removed noise components (which are expected to vary between both time series). Note that the different noise levels actually help to assess the quality of noise removal: Only for completely removed, or suitably suppressed, noise do we

expect a very good comparison of both signals.

We chose to use a rolling linear correlation to present this concept (Fig. 10). Given a filtered time-series and its corresponding indirect one (filter applied before superposition), a linear correlation coefficient ($R^2$) is computed for rolling windows of data, providing time-localized information on the status of noise removal. Dipole E05-01, presented in the upper panel of Fig. 10, shows two distinct signals that are attributed to melt water flow (A/B, to be discussed in detail in section 5.3.3), while dipole

E08-01, presented in the lower panel of Fig. 10, does not exhibit clear signals from temporally localized events. The rolling





linear correlation of Dipole E05-01 with its superposition counterpart is generally close to one, indicating a strong, clear signal. Only one spike to lower correlation values is observed, and can be directly linked to a clear deviation of both signals at this time step (around 5. May). Contrary, while the correlation of dipole E08-01 shows periodic improvements, roughly at daily intervals, the correlation between direct and indirect time series remains relatively low, indicating incomplete removal of high-frequency
noise components.

Please note that the chosen window length of 12.5 hours does slightly distort the SP data, and is chosen here just as a conservative choice to produce a filtered time series with a low share of noise components (Fig. 10a). Also, a low correlation coefficient between direct and indirect signal does not necessarily mean that the given time-series cannot be used for further analysis. We interpret low correlation coefficients merely as an incomplete removal of noise components, whose definition
will vary with specific application goals. For example, it seems that daily variations manifest differently in dipoles E16-E01 and E08-01 (Fig. 10c, black and blue lines), leading to a cyclic variation in correlation (Fig. 10d), and could be caused by different dipole lengths and locations and associated different temperature responses, which in itself could be a valuable target of investigation.

### 5.2.3   Temperature sensors and leakage currents

During the operation of the measurement system, systematic, cyclic jumps were detected in the temperature readings of some of the electrode sensors, but not all of them (Fig. 11a, green dots). The signal has an approximate period of 24 hours, which, at first, indicated diurnal variations of the temperature near the electrodes. However, temperature changes show two roughly discrete levels, which do not fit with natural daily temperature cycles. As a precaution, just before each temperature measurement, the SP-system measures the voltage between two of the terminals (without any current being actively injected). Ideally, no potential
difference should be measured, as the temperature sensor (and supply wires) are passive and isolated components, which only see current flow during excitation for the resistance measurement used to determine the temperature. As such, any measured voltage is an indication of system malfunction, most probably compromised isolation (i.e., electrical contact with the soil, or some kind of short-circuit in the measurement system). Indeed, the observed jumps in temperature coincide with systematic changes in measured voltages (Fig. 11b, blue dots). While we can only speculate on the specific nature of the problem, we
observe that the temperature curve still follows those of neighboring electrodes, with the systematic error superimposing on the actual temperature curve. As such we assume that temperature data can still be extracted (i.e., by averaging or using only those data points with corresponding voltage measurements near 0 V) from the affected temperature sensors.

As a possible explanation we propose leakage currents caused by small short-circuits in the patch panel of the measurement system (marked with red rectangle in Fig. 5, bottom right of the picture). This is supported by the measured temperature of the
data logger, housed next to the connection points (Fig. 11b, red line), which shows temperature lows at times with increased voltage drops over the sensors (small time lags can be explained by the thermal protection of the internal logger temperature sensor with respect to the patch panel). A decrease in temperature can lead to increased condensation, which in turn could lead to a fluid film causing the short-circuit of points of different electrical potential.





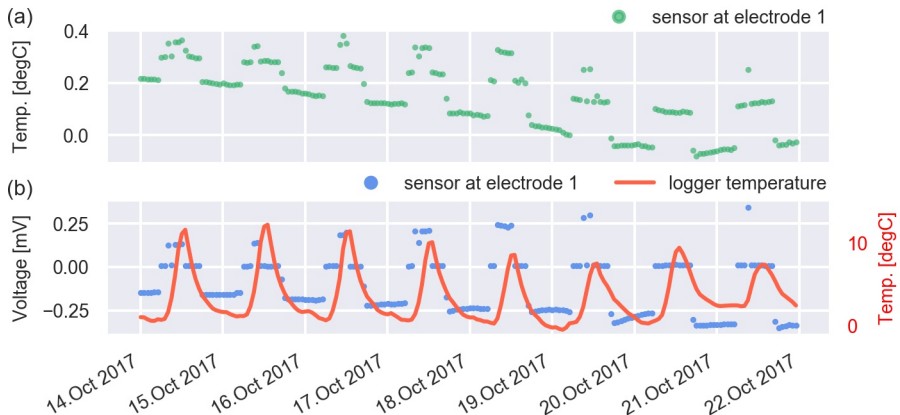

**Figure 11.** a) Temperature curve for sensor at electrodes and 5. b) Corresponding voltages (blue), measured over the Pt100 resistance of the temperature sensor before current injection, and logger temperature (red).

## 5.3 Measurement data

In this section we present selected results of the operating time in 2017 and 2018, with two longer interruptions caused by power issues. These results serve first to show that reliable and consistent data can be measured with the system, and only in secondary role as an attempt at extracting process-related information (which would go beyond the scope of this study).

Temporal and spatial consistency of the data is shortly discussed, as well as two captured SP events in May, 2018, which are interpreted as being caused by snow melt water infiltration near selected electrodes.

### 5.3.1 Long-term SP data

Filtered SP data for the completed measurement time span, starting in September, 2017 and ending in May, 2018 is presented in Fig. 12a, with corresponding electrode temperatures shown in Fig. 12b. In order to visualize long-term trends, a rolling average

filter with a window length of 48.5 hours was applied to the electrical data, suppressing high-frequency noise and pronounced diurnal cyclic signals (c.f. Figs. 10 a, c). The spectral content of the unfiltered data is discussed in section 5.3.4.

The evolution of electrical SP signals is characterized by roughly four phases: A freezing phase (12a-A), which shows both temporal and spatial variations in the SP signals. This period is followed by a "quiet" time, showing only minor variations between dipoles over time (12a-B). After all electrodes reached temperatures below zero degrees, curves do not diverge much,

but show much larger temporal variability (12a-C). Finally, following the data gap in early 2018 (malfunction of solar panel), curves start to diverge again, while maintaining their larger temporal variability (12a-D).

While some electrodes dipped below zero degrees, phase A is still dominated by temperatures around zero, indicating only partially frozen subsoil, in which water flow is still possible. This water flow, coupled to ongoing freezing processes, could lead to streaming potentials and chemical potentials, leading to spatially varying signals.

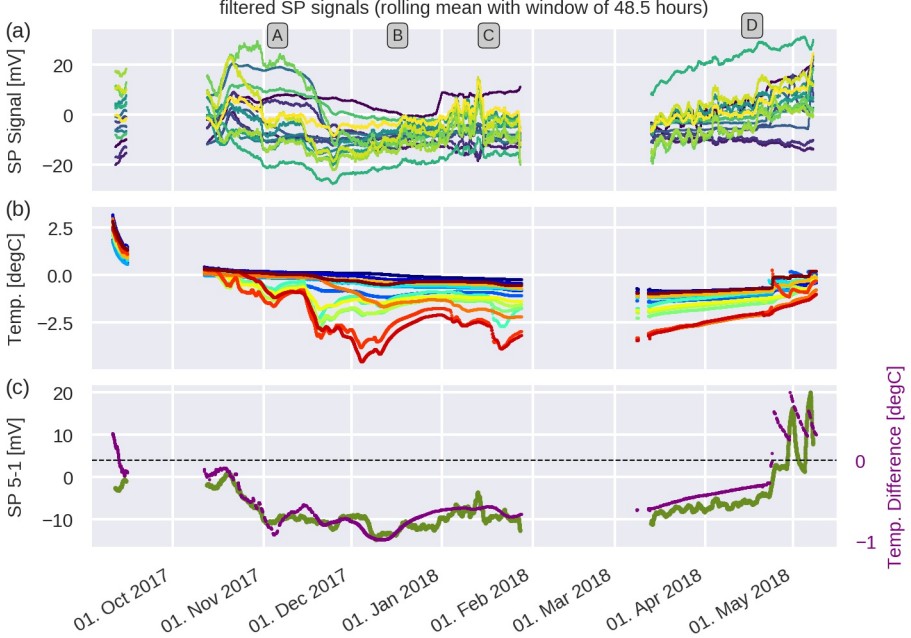

**Figure 12.** SP time series and temperature data. a) All 15 SP time series measured with respect to reference electrode 1. Dipole 2-1 in dark purple, dipole 16-1 in green. Data are filtered with a rolling mean filter (window: 48.5 hours). b) Corresponding temperature curves for all electrodes 1 (blue) to 16 (red). Temperature readings that showed significant voltage drops over the temperature sensors were discarded. c) Correlation of SP dipole 5-1 with electrode temperature difference, $\Delta T_{5-1}$.

Long-term trends of the SP dipoles show a correlation to the temperature differences between the electrodes, with the SP data and temperature curve for dipole 1-5 presented exemplary in Fig. 12c). Note, however, that the observed variation in the electrical signals by far exceeds the temperature coefficients of the electrodes (ca $20\mu V/^\circ C$, Petiau (2000)), suggesting additional processes at play (i.e., highly localized water pockets/movements; geochemical changes). Similar correlations of SP 5 signals to temperature differences at the electrodes were also observed in other settings, e.g. in Friedel et al. (2004). Hu et al. (2020) also suggest changing membrane potentials of the electrodes as sources of measured SP signals.

### 5.3.2 Voltages with respect to duplicate electrodes

Both reference electrodes, with numbers 1 and 16, were duplicated in order to investigate electrode drift and small-scale heterogeneities (Fig. 13 shows temperatures of all four electrodes, as well as the SP voltages between the pairs). As long as all 10 temperatures remain above zero degrees, measured SP voltages fluctuate within 6 mV (Figs. 13a,b, marker A). Afterwards, both voltages stabilize at fixed values, which, however, deviate from zero mV (time span indicated by B). Finally, two temperature events are correlated with transient spikes in the measured voltages (markers C and D).





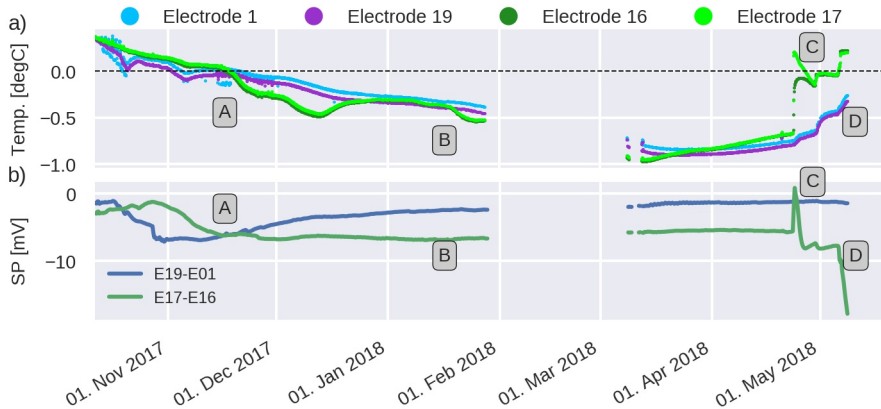

**Figure 13.** Measured voltages between references electrodes 1 and 16 and their respective duplicated electrode, located a few cm away. a) Temperature readings of all four involved electrodes. b) Voltage readings between duplicated electrodes 1 and 19, and 16 and 17.

We interpret the initial phase A as a sign of ongoing hydrological and chemical processes in the soil, with corresponding heterogeneities in chemical and streaming potentials. After freezing (B), a chemical equilibrium ensues and water flow stops, leading to stable voltage readings. The two events (C, D) associated with temperature increases are partially attributed to internal electrode effects, but are also a clear sign of melt water infiltration down to the electrodes. These events are discussed
in the next section.

### 5.3.3   Thawing events

Before operation of the system was interrupted by lightning damage, multiple distinct temperature events that crossed the zero-degree mark were recorded in early May 2018 at some, but not all, electrodes. Each of these temperature events can be associated with changes in the measured SP signals involved with these electrodes, as shortly discussed in the following.

Two of these events were already presented in the last section for electrodes 16 and 17 (Fig. 13a,b - marks C, D). Here, a positive voltage spike is observed at mark C, when only electrode 17 warmed above zero degrees, and a negative, not fully resolved, spike at mark D, when both electrodes, 16 and 17, warmed above zero degrees.

Additional events were found for electrode 5 (Fig. 14b, marks A, B, C). Three fully resolved temperature spikes were observed for this electrode in the end of April, beginning of May, with two large transient SP spikes being observed for
temperature increases above zero degree (Fig. 14c, marks B, C).

In general, sharp increases in electrode temperatures correlate well with transient increases in the corresponding SP readings, although this response is much smaller when both electrodes remain under zero degrees (compare events A and B/C in Fig. 14). At the time of these events a closed snow cover of ca. 1 m was still present at the site (Fig. 14a), strongly indicating energy transport by melt water, which is consistent with previous studies on melt water infiltration into snow covers (e.g., Scherler





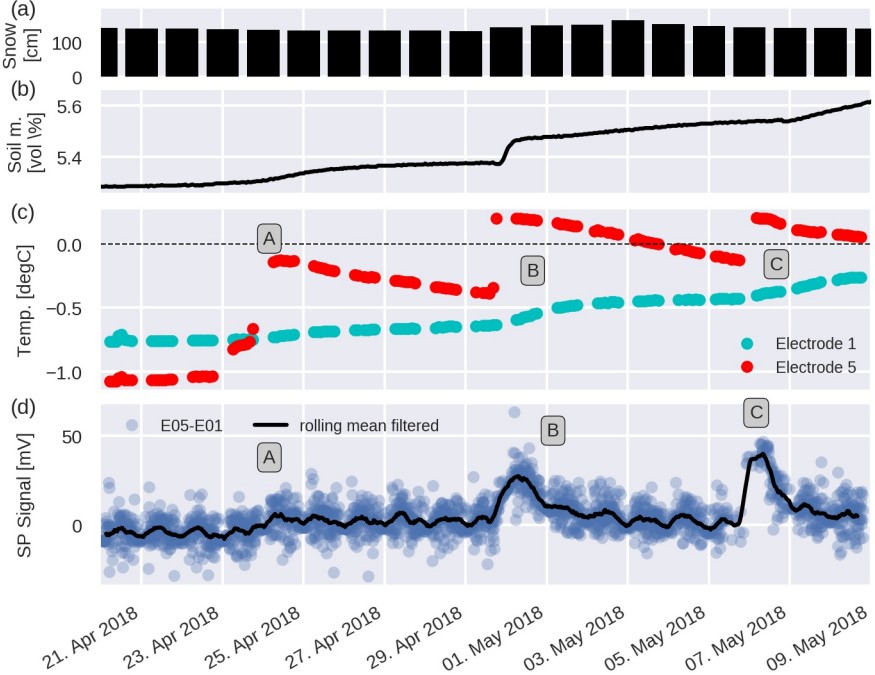

**Figure 14.** SP time-series and associated data over the time window containing three distinct temperature events. a) Daily snow cover height. b) Soil moisture content from nearby PERMOS borehole. c) Temperature curves for electrodes 1 (cyan) and 5 (red). Temperature readings that showed significant voltage drops over the temperature sensors were discarded, leading to the temporal gaps in the curves. d) SP time-series between electrodes 5 and 1. blue: raw data, black filtered data (6.5 hour rolling mean)

et al., 2010; Hilbich et al., 2011). This assumption is supported by increases in soil water content at the time of above-zero events, measured in a nearby borehole (Fig. 14b).

A natural conclusion would be to interpret these transient SP signals as streaming potentials, caused by intermittent melt water flow, consistent with previous findings on early water infiltration for the test cite (e.g., Scherler et al., 2010; Hilbich et al., 2011). Referring back to the laboratory experiment (sect. 5.1.1), we note however that the observed signal is likely comprised also of an additional component caused by geochemical changes in response to the temperature changes.

### 5.3.4 Spectral power analysis

In order to get a first-order idea of spectral components of the data, the irregularly spaced raw data was analyzed using the Lomb-Scargle periodogram (VanderPlas et al., 2012; VanderPlas and Ivezic, 2015). Data from the central data block (10. Nov. 2017 – 28. Jan. 2017) was used, and periods between 1/6 days (4 hours) and 2 days were analyzed (Fig. 15). Data was detrended before analysis, but no further analysis regarding stationarity was done, and would be required if further analysis was attempted.





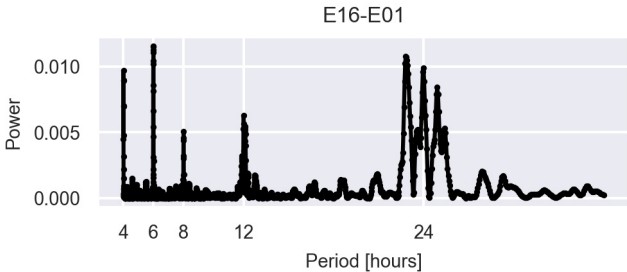

**Figure 15.** Spectral power analysis of E01-E16, using the Lomb-Scargle periodogram. Data was detrended before analysis.

The data exhibits clear diurnal components around the 24 hour period (23, 24, 25 hours). The daily cycles can also clearly be observed in Figs. 10 a, c. Additional signal components can be found for 4, 6, 8, and 12 hour periods. At this point we do not try to attribute any signal sources to those periods, and would suggest further, advanced, time-series analysis before any period other than the 24 hour signal is analyzed due to the inherent non-stationarity of the time-series. However, we would like

to note that all recovered periods can be associated with global tidal processes (e.g., Egbert and Booker, 1992; MacAllister et al., 2016). We can also exclude large influences of the regular ERT measurements conducted at the Schilthorn. While these measurements have a planned measurement interval of 24 hours, measurements only take up to an hour, which would lead to corresponding disturbances in the SP signals for only this duration, and thus cannot explain the observed continuous signals changes over the day (e.g., visible in Figs. 10 a, c).

Elucidating the true nature of any SP signals associated with one or more of these periods is beyond the scope of this study. However, given that the study site remained largely frozen for much of the presented time frame, we can exclude, or at least doubt, some of the explanations given for other studies in respect to our study: Direct temperature-related causes can be excluded due to the absence of clear diurnal temperature signals being recorded at the electrodes. We can also exclude that surface melt water was generated throughout the whole season, given that we only expect significant melt water in winter

due to strong surface radiation caused by direct sunlight, which is often not present in case of cloud covers and limited at the north-facing slope of the Schilthorn.

Blake and Clarke (1999); Kulessa et al. (2003) suggest electrochemical and streaming potentials as possible causes for diurnal variation. However, their potential measurements were explained with variations in electrical fluid conductivity and water pressure caused by periodic pressure and melt-water flows in a summer-time glacier. We do not expect significant

periodic water flow and changes in geochemistry throughout the time of frozen soil at the Schilthorn.

Friedel et al. (2004) discuss magneto-telluric currents, capillary flow in unsaturated media, evaporation, and atmospheric air pressure as additional sources for diurnal, or semi-diurnal, periods. Based on the site characteristics, we see only magneto-telluric signals as possible sources for the signals observed at the Schilthorn, again due to missing flow opportunities in the frozen soil.



## 6   Discussion and Outlook

In this section we shortly discuss the state of the presented measurement system with respect to the previously defined scientific goals, and lay down aspects of the analysis pipeline leading to subsurface flow parameters, incorporating quality checks built into the system. Finally, we offer an outlook on future development opportunities for the presented SP monitoring system.

### 6.1   Monitoring of streaming potentials in Alpine permafrost

Our primary goal is the monitoring of subsurface water flow (generated by melt processes, precipitation, or other means) in alpine permafrost regions. Yet, commissioning a monitoring SP system, such as described in this study, is only one step in a whole pipeline of steps required to produce reliable estimates of water flow, outlined in the following. It was shown that assessment of data quality becomes essential for measurements in such a harsh surrounding, and suitable assessment tools have been discussed to separate signals from noise components. The specific nature of these noise signals will change for different field sites, requiring the establishment of standard operating procedures to guarantee useful SP measurements. Here, it should be noted again that daylight times at the Schilthorn summit shall be considered as a worst case in terms of anthropogenic noise, due to the operation of heavy cable car machinery and associated large ground currents (yet, night-time data shows considerably less noise, still allowing for long-term data monitoring if noise levels overwhelm process signals at daytime). Following the separation of signals and noise, process signals need to be decomposed further into relevant and irrelevant components, separating streaming potentials from all other sources of electrical potentials. One approach would be by isolating certain frequency bands known to contain more or less periodic water flow information, or to select transient signals in response to additional environmental data, such as borehole data, precipitation and other weather measurements (in our case provided by the PERMOS network and the Federal Office of Meteorology and Climatology MeteoSwiss).

On the other hand, if the nature of 'noise' processes involved is known, they often can be modeled, and therefore their contribution can be subtracted from a given measurement signal. This is a common procedure, for example, to eliminate telluric signal components (e.g., Perrier et al., 1997), or to account for systematic changes in SP signals due to topography (the 'topographic effect', e.g. discussed in Scapozza et al., 2008). Hu et al. (2020) even attempt to model not only process signals, but also electrode effects over time.

Having isolated the SP signals attributed to subsurface water flow, actual flow parameters need to be derived. Analysis can be done either using the Helmholtz-Smoluchowski equation, or by considering the excess charges at the EDL, and their dynamics under water flow (e.g., Jougnot et al., 2020). In any case, usually the coupling coefficient is determined from complementary laboratory information.

Finally, the streaming potentials need to be analyzed not only with regard to time, but also with regard to space, given the large surface and subsurface heterogeneities in mountainous areas. Here, methods ranging from simple to advanced could be used, correlating the orientation of specific dipoles to water flow (simple), or applying advanced tomographic algorithms to produce 3D subsurface water flow patterns from measured SP signals (e.g., Jardani et al., 2007; Ahmed et al., 2013, 2020).





As such, we believe that the presented system offers huge potential for the monitoring of water flow in permafrost regions and can be replicated with relative ease to allow a wide application of the method. However, future efforts should be focused on integrating SP measurements in the established monitoring networks, such as PERMOS, in order to provide complementary data required for the analysis outlined above.

## 6.2 Possible future extensions and development directions

At this point the system is capable of monitoring SP signatures and electrode temperatures. While additional environmental data is provided for the Schilthorn site by the PERMOS network station located right next to our system, future extensions should integrate the measurements of all relevant data required to analyze SP signatures with regard to subsurface flow. The additional, integrated measurement of snow cover, fluid pressure, salinity and pH, and subsurface electrical conductivity greatly

enhance the analysis possibilities, and are required if subsurface tomography is to be implemented. Also, contact resistances between electrodes should be measured for the same dipoles as used for the SP readings (at this point, for technical reasons this is not the case), allowing better analysis of long-term changes. Contact resistances should optimally be measured using alternating currents, thereby reducing electrode polarization and increasing accuracy of the readings. Aside from the need to improve electrical isolation in the system, the measurement of voltage drops over temperature sensors provides a useful quality

assessment tool that can help in identifying periodic signals caused by systematic measurement errors.

On the technical side, one obvious shortcoming of the system is its reliance on solar power to charge the batteries. While this solution is cheap, small, and suitable for rough terrain, the use of fuel cells can greatly enhance available power and its reliability. Adding more available power to the system would also allow for higher measurement frequencies, which would facilitate the capture of faster flow events, improve noise monitoring, and reduce or fully suppress aliasing caused by the

large cut-off frequency of the filters currently implemented. The dependence of the low-pass filter on the soil resistance could be reduced by using a high-impedance input, low-impedance output operational amplifier buffer, which would also reduce effects of the input resistance of the data logger. For example, Zimmermann et al. (2008) realized an input impedance of approximately 500 $G\Omega$ for active electrical potential measurements with such a setup. Finally, at this point we refrained from implementing a bi-directional data exchange with the logging system, mainly due to power deliberations. This turned out to be

much more constraining than previously thought, given that we were unable to reprogram the logger system in response to the data received. Future changes to the system will focus on establishing bi-directional communication.

The design and implementation of the presented SP monitoring system is based on our experience with established classical geophysical devices. Yet, the recent emergence of the so-called internet of things (IoT) also lead to a large number of technical advances that can be easily envisioned to be of use for geophysical applications, especially in niche fields such as the one

discussed in this study. Prominent examples of novel sensor networks are the Xsense and PermaSense network (Beutel et al., 2011; Weber et al., 2019). High quality data acquisition and processing units are being produced in huge quantities, driving down costs for design and implementation. Micro-computer based environments around the Arduino or ESP32 environments allow fast and cheap development of targeted hardware, while facilitating the large consumer-orientated development environments. 3D printing allows researchers to easily design custom casings, and freely available software libraries speed up





development of the corresponding firmware and backend software. In addition, we see the increasing demand for low-power connectivity between IoT devices as the most effective development for geophysical applications, having lead to the creation of connectivity technologies such as LoRa and SigFox, which offer low-bitrate connections over very long ranges (km range), while using much less power than established wireless technologies, such as Wifi or GSM/LTE. As such, we envision a move

from centralized measurement setups, based around one expensive logger system, towards a distributed system of individual, connected, low-cost measurement units. SP signatures could then be measured by small systems with only few connected electrodes, increasing resilience towards disruptions by the elements, increasing area coverage without requiring very long cable connections, and finally driving down deployment and maintenance costs. One small downside of such a system is that the distributed nature also implied multiple reference potentials toward which measurements are conducted, requiring additional

processing steps (e.g., Minsley et al., 2008).

Measurements of environmental data in alpine environments are commonly prone to be disrupted by various events, such as power failures and lightning damage, and thus result in irregular spaced time-series. If precise frequency analysis of the measured data is required, for example, in order to extract or identify relevant signatures concentrated on certain frequency bands, then advanced analysis techniques must be applied to irregular time-series, such as the Lomb-Scargle periodogram

(VanderPlas et al., 2012; VanderPlas and Ivezic, 2015), or improved multi-taper spectral estimation methods (e.g., Chave, 2019).

## 7 Conclusions

In this study we present a new design for a long-term SP and temperature monitoring system aimed at measurements in remote, alpine permafrost regions. The system was installed at the Schilthorn summit, and operational capabilities were tested over

roughly 7 months, starting in September, 2017. Preliminary data analysis emphasizes the importance of quality assessment of the captured data, which show significant noise contributions, of which a large portion is attributed to anthropogenic activities at the mountain top. The data analysis includes checks for connectivity of electrodes and temperature sensors, but has also built-in redundancy in the design of electrical measurement dipoles. This redundancy is not only useful in case of problems with a reference electrode, but it also allows statistical checks to be run on time-series extracted from the measured raw data.

The system minimizes the use of custom electrical circuits, and, where possible, uses commercially available equipment in order to facilitate easy duplication throughout the community. Preliminary analysis of captured data indicates a sensitivity to the state of frozen/unfrozen subsurface, and multiple captured thermal events indicate a sensitivity to melt water flow at early times of the year. Yet, it is important to see the self-potential method not as a stand-alone tool, but as an important addition to the toolbox of geophysical methods, which will thrive best in multi-geophysical measurement environments. As such we

believe that the presented system provides a solid base for scientific analysis of subsurface water flow by means of electrical signatures, be it in (partially) frozen soil in and before winter, or unfrozen conditions in the summer. Information on subsurface water flow thus gained will provide important information that can potentially be integrated into risk assessment frameworks for mountainous regions.



*Code and data availability.* All data and Python scripts used to created the figures in the study will be made available using zenodo.org

*Author contributions.* AK initiated research in the topic and provided funding and resources as well as general supervision. AK, MW conceptualized the research with support on laboratory and field concepts from FMW and JKL. MW designed the instrumental setup, data acquisition and data processing procedures. MW, FMW, JKL and AK performed the initial field installation and subsequent field campaigns.

5 Christin H and Christian H provided resources, auxiliary data and logistical support. All authors discussed and interpreted the results. MW, AK, and FMW discussed the scope and structure of the manuscript. MW prepared the manuscript with contributions from all authors.

*Competing interests.* The authors declare that they have no conflict of interest.

*Acknowledgements.* The authors would like to thank Schilthornbahn AG for logistical support. Environmental data was provided by the PERMOS network and MeteoSwiss. Dirk Handwerk and Henrik Blanchard provided valuable help in constructing the system. Egon Zim-

10 mermann provided important input regarding the implementation of electrical details. We are grateful to David Sciboz, Cécile Pellet and Coline Mollaret for on-site help with field work and maintenance.



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
