# Peer review of "A monitoring system for spatiotemporal electrical self-potential measurements in cryospheric environments"

_Geoscientific Instrumentation, Methods and Data Systems, 2020_

## Short Comment (SC1) · 17 Mar 2020

As an actual SP user to study hydrological processes in the critical zone, I also think that there is a growing need to provide a state of the art equipment for "long-term, year round, unsupervised operation [which] must be ensured to minimize human intervention" (and, from my point of view, not restricted to permafrost regions). In general, I think that the proposed system is carefully designed and well-though for optimal long-term measurements, using state-of-the-art knowledge about both electrode, electronics, and SP theory. I particularly like that a good care is taken for monitoring the temperature exactly where electrodes are in contact with the ground and the careful

study of this crucial effect in such extreme environment.

I went through this nice manuscript in open discussion and I have some questions/comments below. That said, I really like this manuscript and the really nice system that you designed. I'm also quite eager to read the process-based study/paper that will follow this publication.

Best,

Damien Jougnot

CNRS scientist, Sorbonne University

Page 3 lines 18-23: It seems that HS equation is working fine for any kind of geometry (it was also demonstrated around large enough spheres, i.e. in colloid sciences) as long as the surface conductivity can be neglected. We try to discuss and highlight this in our recent paper Jougnot et al. (2019).

P.4 lines 8-20: one can note that Doussan et al. (2002) did perform some long-term monitorings of vertical flow and Voytek et al. (2019) vertical and horizontal close to a tree.

P.8 line 3: "vertical" could be misleading as it could be understood as at various depths.

P.9 line 10: why is fig. 6 called before fig. 4 ?

P.9 lines 11-16: It is good to make measurements with respect to a single reference (i.e., total field instead of dipoles) and having a duplicate of that seems like a good idea, but I am a bit confused by this paragraph. Could the author elaborate on the electrical circuit wiring they propose to be able to measure it with respect to two different references ? is it at the same time ? alternatively ?

P.13 Fig. 7: This experiment is really interesting and changes are bigger than I would have expected. What about between standing electrodes ? And did the authors check between both pairs of buried vs. standing or only one ?

P.14 Section 5.1.2: Given the dynamic of processes sensed by SP signal, is 1h rolling-mean a good choice ? In Jougnot et al. (2015) and Hu et al. (2020) we used 5 min windows and probably missed/were affected by higher frequency SP signal.

P.23 section 6.1: Something which is often done when conducting a SP mapping/profile (and I think should be done), is to remove the reference used for the measurement and re-reference the signal. Could be interesting to show/discuss this point.

P.23 section 6.2: I strongly encourage the author to include dielectric permittivity and local electrical conductivity measurements to their system.

P.24 line 23: I don't think G should be in italics.

References:

Doussan, C., Jouniaux, L., & Thony, J. L. (2002). Variations of self-potential and unsaturated water flow with time in sandy loam and clay loam soils. Journal of Hydrology, 267(3-4), 173-185. Jougnot, D., Mendieta, A., Leroy, P., & Maineult, A. (2019). Exploring the effect of the pore size distribution on the streaming potential generation in saturated porous media, insight from pore network simulations. Journal of Geophysical Research: Solid Earth, 124(6), 5315-5335. Jougnot, D., Linde, N., Haarder, E. B., & Looms, M. C. (2015). Monitoring of saline tracer movement with vertically distributed self-potential measurements at the HOBE agricultural test site, Voulund, Denmark. Journal of Hydrology, 521, 314-327. Voytek, E. B., Barnard, H. R., Jougnot, D., & Singha, K. (2019). Transpiration‐and precipitation‐induced subsurface water flow observed using the self‐potential method. Hydrological Processes, 33(13), 1784-1801.
* * *

---

## Referee Comment (RC1) · Anonymous Referee #1 · 9 Apr 2020

Being involved in several long-term magnetotelluric monitoring, I am really interested in the topic faced in the manuscript. As a general comment, I think that the authors made an excellent job in designing and realizing an effective monitoring system. Without entering too much in the discussion of the quality of the collected data whose analysis goes beyond the aim of the presented paper, I have just few comments/questions (see below).

In figure 2, the electrode layout at the Schilthorn summit is presented. From the colour scale adopted, it is only clear that the S-E corner (upper left corner of the figure) of the layout is where the maximum elevation of the area is reached but it is not clear the

topography of the area. The Authors could try to make clearer the figure by adding isolines or by changing the used colour scale.

Page 6, lines 1-3: the sentence "Past studies have also greatly benefited from magnetic measurements for assessment of magneto-telluric signal components, water pressure sensors (e.g., Blake and Clarke, 1999), and pH probes." is quite confusing. The simultaneous record of magnetic signals allows the application of the magnetotelluric method which provides information of the electrical subsoil structure at a much higher depth of the ERT. Are the benefits to which the Authors refer related to a deeper investigation depth of the magnetotelluric?

Page 14, lines 8-9: "The observed signal levels exhibited are large in comparison 10 to expected low-frequency SP signals, yet low-pass filtering reveals a clear diurnal signal (Fig. 2, red line)." I think there is a mistake, the correct figure is FIG. 8

Figure 9: The figure shows the temporal evolution of contact resistances over the whole measurement period of 2017 and 2018. Commenting this figure, the Authors state at pag 14 that "However, the actual values of the resistances differ with the electrode pairs, with very high resistances (red colours) occurring for some electrode pairs, and 30 relatively low resistances (blue) showing up intermittently for the electrode pairs 3-4 and 6-7." By looking at figure 9, it seems that the pairs 3-4 and 6-7 undergo to an abrupt an simultaneous change in contact resistance at the end of November (?) but a similar change is not observed in other pairs involving electrodes 3,4,6,7. This last observation seems to exclude an electrode malfunction. How do the authors interpret this phenomenon? Is in their opinion linked to a variation of the measuring condition?

Figure 12: By looking at the temperature reading related to each electrode position, it seems strange that 2 electrodes (I am assuming the 15 and the 16) have a temperature which is more than 2°C lower than the temperature recorded in other electrode positions. Considering the relatively small distances between the electrodes, is this difference realistic? Furthermore, figure 13a reports again the temperature reading of the

[Figure]

electrode 16 and the temporal trend observable here is different by the one reported in figure 12. Am I interpreting in a wrong way one of the figure or is there something wrong?

Page 22, lines 4-6: "However, we would like to note that all recovered periods can be associated with global tidal processes (e.g., Egbert and Booker, 1992; MacAllister et al., 2016)." A simpler explanation involves the presence of a diurnal variation of geomagnetic field which results from perturbations of the Earth's ionosphere and depends to disturbances of the upper-atmosphere, mainly due to solar activities. As also reported in some magnetotelluric textbook (e.g. Chave AD, and Jones AG (2012). The magnetotelluric method: Theory and practice. Cambridge: Cambridge University Press.) this diurnal variation is expected to affect also the telluric recordings.
* * *

---

## Referee Comment (RC2) · Anonymous Referee #2 · 24 Apr 2020

As a user of SP field data, this manuscript is of great interest. There is a growing body of work on the mechanisms of SP generation, but still an overall lack of understanding of how to collect and process SP data appropriately, particularly when it comes to the collection of long-duration data sets.

I agree with the authors, and the review of Damien Jougnot, that there is a need to develop equipment for long-term measurements. I would add that along with collecting the data, there is a need to understand the various sources of noise in SP data and how they can be treated. While the authors have taken great care to point out that they are not making process-based interpretations of the SP data in this instrumentation paper,

the scale of the various noises identified brings into question whether "extraction of robust data" is possible. Most of the noise sources are on the same scale of magnitude as the signals themselves, if not larger. In the discussions, multiple possible noise sources are presented, without thorough evaluation of how to discriminate one from another.

Specific comments: P9 L11: It would be useful to have information about the specific logger is used, before detailing the sampling schedule.

P9 L11-16: I agree with SC1, this text is unclear. Could you provide I diagram that shows which pairs are measured?

P11 L1: Seems worth noting that this was installed due to actual lightning damages, mentioned in passing on P20.

P13 F7: The vertical black line is missing.

P13 F7 and P20 F13: These two figures make it very clear that it is necessary to 'get back to basics' regarding SP measurements. This magnitude signal noise, measured between electrodes spaced at a cm scale, brings into question the ability to make interpretations on SP measurements of the same magnitude, collected between electrodes meters apart. This is motivation for the community to work on this persistent problem, but would be a great place to add some discussion of how this might be tackled.

P13 F8: Consider plotting the red line on the same scale as the original data, and then providing a second plot of it at a larger scale. It's challenging to interpret when the 0 of the two graphs is in different places on the two axes. Caption says 2-hour average, legend say 1-hour. Is it possible this noise is from the solar panels?

P16 F10: Make the two curves that you are comparing E05-E01, and E05-01 superposition the strongest curves (non-black?), and the E16-E01 curve fainter. I was visually comparing blue and purple for a long time, until I realized there was a black line, too. As other reviewers have mentioned, a moving average of 12.5 hours seems very long,

if you are aiming to measure subsurface water flow, which might occur on shorter time scales.

P19 F12: There are at least six different greens. Yellow? Regardless, consider correlating the color of the measured pair, and the non-reference electrode. Alternatively, since there is no way to differentiate in the 15-16 individual curves, consider plotting all data as faint/thin black lines and only highlighting in color the few that you discuss in more detail.

P20 L12: What is meant by resolved in this case? Solved or completely shown?

P24 L1: Remove "huge" potential. A great tool to collect these data has been presented, but many questions persist about the use of the data.

In conclusion, the authors present a very detailed information about how they addressed the numerous challenges of working in such a remote environment. With some work on the organization of ideas, the manuscript could be of interest to many who are considering adding SP to ongoing research in any environment.

---

## Author Comment (AC1) · 10 Jun 2020

**Responses to comments for manuscript gi-2020-5-RC1**

Dear editor, dear authors,

thank you very much for the encouraging reviews and comments. We are convinced that the manuscript was much improved by incorporating the responses to your comments and hope that we accomplished this to your satisfaction.

In the following you will find detailed answers to the comments SC1, RC1.

Best regards

Maximilian Weigand, Florian Wagner, Jonas Limbrock, Christin Hilbich, Christian Hauck, Andreas Kemna

**Responses to gi-2020-5-SC1 by Damien Jougnot**

> Damien Jougnot:  As an actual SP user to study hydrological processes in the critical zone, I also think that there is a growing need to provide a state of the art equipment for "long-term,year round, unsupervised operation [which] must be ensured to minimize human intervention" (and, from my point of view, not restricted to permafrost regions). In general, I think that the proposed system is carefully designed and well-though for optimal long-term measurements, using state-of-the-art knowledge about both electrode, electronics, and SP theory. I particularly like that a good care is taken for monitoring the temperature exactly where electrodes are in contact with the ground and the careful study of this crucial effect in such extreme environment. I went through this nice manuscript in open discussion and I have some questions/comments below. That said, I really like this manuscript and the really nice system that you designed. I'm also quite eager to read the process-based study/paper that will follow this publication.
> Best,
> Damien Jougnot
> CNRS scientist, Sorbonne University

First of all we would like to thank you very much for this constructive voluntary review of our manuscript. We also hope that the presented work can prove to be useful outside the permafrost scope. Therefore we now explicitly mention this in the conclusion.

> Point 1:  Page 3 lines 18-23: It seems that HS equation is working fine for any kind of geometry (it was also demonstrated around large enough spheres, i.e. in colloid sciences) as long as the surface conductivity can be neglected. We try to discuss and highlight this in our recent paper Jougnot et al. (2019).P.4

Good point. We changed the corresponding sentence and included your reference.

> Point 2:  lines 8-20: one can note that Doussan et al. (2002) did perform some long-term monitorings of vertical flow and Voytek et al. (2019) vertical and horizontal close to a tree.

We slightly restructued the citations for the past SP studies and included Doussan et al 2002. Voytek et al 2019 was already mentioned, but is now explicitly mentioned as a long-term study.

> Point 3: *P.8 line 3: "vertical" could be misleading as it could be understood as at various depths.*

Agreed. We changed the sentence to read:"The SP setup consists of 20 unpolarizable *Pb/PbCl$_2$* SP electrodes of the Petiau-type (Petiau, 2000), arranged in a 4 x 4 grid of 16 electrodes on the surface with an approximate distance of 7.5 m between rows and columns (Fig. 2)."

> Point 4: *P.9 line 10: why is fig. 6 called before fig. 4 ?*

Showing just the electrode with the temperature sensor attached was, in our opinion, not important enough to add a stand-alone figure. However, we agree that forward-referencing a figure is not good practice and could confuse the reader. Therefore we removed the reference to Fig. 6 at this location.

> Point 5: *P.9 lines 11-16: It is good to make measurements with respect to a single reference(i.e., total field instead of dipoles) and having a duplicate of that seems like a good idea, but I am a bit confused by this paragraph. Could the author elaborate on the electrical circuit wiring they propose to be able to measure it with respect to two different references? Is it at the same time ? Alternatively ?*

This issue was also brought up by referee #2. We reworked this specific paragraph and go into more detail regarding the multiplexing involved (there are multiple levels of multiplexing going on: first, a hard-coded multiplexing done using a routing board, and then an internal multiplexing within the logger, which measures sequentially). We additionally reworked the schematic that depicts the electrical routing of the multiplexer (routing) board (Fig. S.5) and added a reference in the main text to this figure.

> Point 6: *P.13 Fig. 7: This experiment is really interesting and changes are bigger than I would have expected. What about between standing electrodes? And did the authors check between both pairs of buried vs. standing or only one ?*

Yes, we measured these dipoles two, but did not include them in Fig. 7 for the sake of clarity. In response to your comment we added those dipoles to the supplement as Fig. S9.

While there are some variations in the strength and clarity of the responses, especially for the freezing events, the main message remains: dipoles between standing and buried electrode exhibit larger signature changes than those between electrodes with similar thermal characteristics.

> **Point 7**: *P.14 Section 5.1.2: Given the dynamic of processes sensed by SP signal, is 1h rolling-mean a good choice? In Jougnot et al. (2015) and Hu et al. (2020) we used 5 min windows and probably missed/were affected by higher frequency SP signal.*

No, the rolling-mean (actually we used a 2-hour rolling mean) probably is not a good choice for certain SP processes (please also see our responses to the other reviewers). Yet, some processes that we are interested in (for example thawing/freezing) show long-enough signals to not be disturbed much by our choice of window length (as can be seen in Fig. 10, for example). As such for further analysis of our (and future) data, it will be important to select process targets that can be captured by the system (i.e., measurement intervals) in the given noise environment (saturation of certain frequency bands).

As stated in the paper, at this point we retain from changing the selected window length as the process analysis is out of scope for the manuscript. Please note that we discuss the choice of filters in Sect. 5.2.2, starting in line 7.

> **Point 8**: *P.23 section 6.1: Something which is often done when conducting a SP mapping/profile (and I think should be done), is to remove the reference used for the measurement and re-reference the signal. Could be interesting to show/discuss this point.*

If I'm not mistaken, this re-referencing commonly refers to measurements done with different reference electrodes (i.e., moving dipoles). As we only measure with respect to one reference electrode (no. 1), there is no possibility to remove the reference here without additional electrodes/measurements.

However, as we also measure with respect to electrode 16, section 5.2.2 does exactly the type of re-referencing that you refer to, effectively eliminating the reference electrode nr 16 from the measurements, and re-referencing with respect to electrode 1. We added a sentence to Sect. 5.2.2 to mention that the procedure of superposition is effectively a re-referencing.

> **Point 9**: *P.23 section 6.2: I strongly encourage the author to include dielectric permittivity and local electrical conductivity measurements to their system.*

Added to the discussion.

> **Point 10**: *P.24 line 23: I don't think G should be in italics.*

Fixed.

**References**

Petiau, G.: Second Generation of Lead-lead Chloride Electrodes for Geophysical Applications, Pure and Applied Geophysics, 157, 357–382, doi: 10.1007/s000240050004, 2000.

**Responses to gi-2020-5-RC1 by Referee 1**

> Intro*: Being involved in several long-term magnetotelluric monitoring, I am really interested in the topic faced in the manuscript. As a general comment, I think that the authors made an excellent job in designing and realizing an effective monitoring system. Without entering too much in the discussion of the quality of the collected data whose analysis goes beyond the aim of the presented paper, I have just few comments/questions (see below).*

Thank you very much! Indeed, as also noted by reviewer number 2 (RC2) much remains to be analyzed with regard to the captured data, or future measurements on this site, and with specific process-based analysis.

> Point 1*: In figure 2, the electrode layout at the Schilthorn summit is presented. From the colour scale adopted, it is only clear that the S-E corner (upper left corner of the figure) of the layout is where the maximum elevation of the area is reached but it is not clear the topography of the area. The Authors could try to make clearer the figure by adding isolines or by changing the used colour scale.*

We added isolines in 1 m increments to the figure, as well as improved the readability of the electrode numbers.

> Point 2*: Page 6, lines 1-3: the sentence "Past studies have also greatly benefited from magnetic measurements for assessment of magneto-telluric signal components, water pressure sensors (e.g., Blake and Clarke, 1999), and pH probes." is quite confusing. The simultaneous record of magnetic signals allows the application of the magnetotelluric method which provides information of the electrical subsoil structure at a much higher depth of the ERT. Are the benefits to which the Authors refer related to a deeper investigation depth of the magnetotelluric?*

No, we are referring to signal components in the measured SP signals that originate from telluric currents in the subsurface. For our purposes these telluric signals are noise and could be removed by the concurrent measurement of the magnetic field. For example, this was done in Perrier et al. (1997), Sec. 3.2.

 We slightly changed the sentence to clarify that point.

> Point 3: *Page 14, lines 8-9: "The observed signal levels exhibited are large in comparison to expected low-frequency SP signals, yet low-pass filtering reveals a clear diurnal signal(Fig. 2, red line)." I think there is a mistake, the correct figure is FIG. 8*

Fixed.

> Point 4: *Figure 9: The figure shows the temporal evolution of contact resistances over the whole measurement period of 2017 and 2018. Commenting this figure, the Authors state at page 14 that "However, the actual values of the resistances differ with the electrode pairs, with very high resistances (red colours) occurring for some electrode pairs, and relatively low resistances (blue) showing up intermittently for the electrode pairs 3-4 and 6-7." By looking at figure 9, it seems that the pairs 3-4 and 6-7 undergo to an abrupt an simultaneous change in contact resistance at the end of November (?) but a similar change is not observed in other pairs involving electrodes 3,4,6,7. This last observation seems to exclude an electrode malfunction. How do the authors interpret this phenomenon? Is in their opinion linked to a variation of the measuring condition?*

We suppose that this is an effect of localized subsurface freezing, which would increase the subsurface resistance to a point where it significantly changes the contact resistance measurements (the common assumption that the contact resistance measurements are dominated by the electrode-soil interface does not account for the large resistance values and changes caused by subsurface freezing).
  We amended the corresponding text paragraph in sect. 5.2.1 accordingly.

> Point 5: *Figure 12: By looking at the temperature reading related to each electrode position, it seems strange that 2 electrodes (I am assuming the 15 and the 16) have a temperature which is more than 2 C lower than the temperature recorded in other electrode positions. Considering the relatively small distances between the electrodes, is this difference realistic? Furthermore, figure 13a reports again the temperature reading of the electrode 16 and the temporal trend observable here is different by the one reported in figure 12. Am I interpreting in a wrong way one of the figure or is there something wrong?*

A temperature difference of 2 degrees between electrodes on small scales is not uncommon at this altitude, possibly caused by microtopography and variation in snow cover (less insulation against the much colder air temperatures in winter) at these electrodes. Normal snow cover height in winter at Schilthorn is well above 1.50 m, which completely decouples the subsurface temperatures from the atmospheric temperatures. If at two electrodes the snow is, say, only 0.5 m thick, then they are much more cooled down by cold air temperatures over the winter period.

Regarding the temperature curves in Figs. 12 and 13: The dominant red temperature curve in Fig. 12b actually belongs to electrode 15. There is an even darker red curve lying close to the 0 degree mark, and this one belongs to electrode 16 and fits with Fig. 13.

> Point 6: *Page 22, lines 4-6: "However, we would like to note that all recovered periods can be associated with global tidal processes (e.g., Egbert and Booker, 1992; MacAllister et al., 2016)." A simpler explanation involves the presence of a diurnal variation of geomagnetic field which results from perturbations of the Earth's ionosphere and depends to disturbances of the upper-atmosphere, mainly due to solar activities. As also reported in some magnetotelluric textbook (e.g. Chave AD, and Jones AG (2012).The magnetotelluric method: Theory and practice. Cambridge: Cambridge UniversityPress.) this diurnal variation is expected to affect also the telluric recordings.*

Thanks for this information, we added a corresponding sentence with references to this section of the manuscript.

**References**

Perrier, F., Petiau, G., Clerc, G., Bogorodsky, V., Erkul, E., Jouniaux, L., Lesmes, D., Macnae, J., Meunier, J., Morgan, D., Nascimento, D., Oettinger, G., Schwarz, G., Toh, H., Valiant, M., Vozoff, K., and Yazici-Cakin, O.: A One-Year Systematic Study of Electrodes for Long Period Measurements of the Electric Field in Geophysical Environments, Journal of geomagnetism and geoelectricity, 49, 1677–1696, doi: 10.5636/jgg.49.1677, 1997.

**Responses to gi-2020-5-RC2 by Referee 2**

> Introduction: *As a user of SP field data, this manuscript is of great interest. There is a growing body of work on the mechanisms of SP generation, but still an overall lack of understanding of how to collect and process SP data appropriately, particularly when it comes to the collection of long-duration data sets. I agree with the authors, and the review of Damien Jougnot, that there is a need to develop equipment for long-term measurements.*

Thanks for the encouraging words.

> Point 1: *I would add that along with collecting the data, there is a need to understand the various sources of noise in SP data and how they can be treated. While the authors have taken great care to point out that they are not making process-based interpretations of the SP data in this instrumentation paper, the scale of the various noises identified brings into question whether "extraction of robust data" is possible. Most of the noise sources are on the same scale of magnitude as the signals themselves, if not larger. In the discussions, multiple possible noise sources are presented, without thorough evaluation of how to discriminate one from another.*

We fully agree with the reviewer that more work needs to be directed at identifying and removing the different noise sources. However, as discussed in the manuscript the categorization of signals sources as noise is highly problem specific and therefore must be closely coupled to the understanding of any processes to be analyzed.

We also like to point out that the large magnitudes of random and systematic noises do not necessarily prevent the extraction of process signals. Provided that the noise does not saturate the process signal in the frequency domain, corresponding band-pass filters could be used to remove most of the random noise. If the nature of the noise processes is known (e.g., electrode effects), a numerical modeling can be attempted to then subtract the noise signature from the measurement signal. This type of noise removal will, however, require the careful co-analysis of multiple different measurement parameters and cannot be attempted by using SP measurements alone.

Finally, we believe that consistency checks such as presented in Sect. 5.2.2 will help researchers to strengthen their confidence in their data.

These points are discussed in Sect. 6.1 - we tried to sharpen the points you make using minor edits.

As with any geophysical measurement, you are correct in stating that if the noise components dominate the signals even after all cleaning procedures were applied, no further process-related information can be reliably extracted.

> **Point 2**: *P9 L11: It would be useful to have information about the specific logger is used, before detailing the sampling schedule.*

This information was added, along with a note that the data recording equipment is discussed in detail below.

> **Point 3**: *P9 L11-16: I agree with SC1, this text is unclear. Could you provide I diagram that shows which pairs are measured?*

(Same reply as to the corresponding issue of SC1) We reworked this specified paragraph and go into more detail regarding the multiplexing involved (there are multiple levels of multiplexing going on: first, a hard-coded multiplexing done using a routing board, and then an internal multiplexing within the logger, which measures sequentially). We additionally reworked the schematic that depicts the electrical routing of the multiplexer (routing) board (Fig. S.5) and added a reference in the main text to this figure.

> **Point 4**: *P11 L1: Seems worth noting that this was installed due to actual lightning damages,mentioned in passing on P20*

This information was added while also addressing the previous issue

> **Point 5**: *P13 F7: The vertical black line is missing.*

Fixed.

> **Point 6**: *P13 F7 and P20 F13: These two figures make it very clear that it is necessary to 'get back to basics' regarding SP measurements. This magnitude signal noise, measured between electrodes spaced at a cm scale, brings into question the ability to make interpretations on SP measurements of the same magnitude, collected between electrodes meters apart. This is motivation for the community to work on this persistent problem, but would be a great place to add some discussion of how this might be tackled*

We agree that these issues need to be dealt with, at some point. However, we would like to note that quite a lot of studies were able to extract reliable information from their SP signatures, despite

the possibility of any effects such as presented here. As such these effects cannot be present at all times.

In addition, both figures, Fig. 7 and Fig. 13 show SP effects in response to a certain process, although not necessarily the expected ones: Fig. 7 shows the effects of inhomogeneous warming of electrodes (as far as our hypothesis goes) **in response** to a freezing/thawing process, while Fig. 10 shows signatures in response to heat/fluid flow effects, and in themselves are therefore important indicators for the processes involved.

Also, it has yet to been seen how large the negative effects are in comparison to larger-scale flow signatures. Fig. 7 does not exhibit any fluid flow and shows data from an laboratory experiment with rather large and fast temperature gradients, while Fig. 10 does (supposedly) only show signatures from localized fluid flow.

We added some clarifying sentences to sect. 5.1.1 in which we point out the rather large temperature gradient in the laboratory experiment.

We also added a paragraph to the discussion (Sect. 6.1, p. 24, lines 27+) in which we discuss that temporal and spatial scales of the processes involved must be taken into account, and subsequently for each process scale one must decide if the recorded data should be further analyzed.

> Point 7: *P13 F8: Consider plotting the red line on the same scale as the original data, and then providing a second plot of it at a larger scale. It's challenging to interpret when one of the two graphs is in different places on the two axes. Caption says 2-hour average ,legend say 1-hour. Is it possible this noise is from the solar panels?*

We added another subplot with the zoomed-in filtered curve, as you suggested.

It is very unlikely that the noise originates in the solar panel for different reasons: First, we are not aware of any intrinsic noise process within the solar cells that could somehow be transferred to the logger and have never seen any systematic differences when operating the system with solar power or only with batteries. Second, systematic fluctuations would need to originate in incoming solar radiation, which we are fairly sure did not fluctuate that fast and with such a regularity, as observed in the data. Please also see Fig. S.8 in the supplement for a time-frequency wavelet analysis of the noise signals.

> Point 8: *P16 F10: Make the two curves that you are comparing E05-E01, and E05-01 superposition the strongest curves (non-black?), and the E16-E01 curve fainter. I was visually comparing blue and purple for a long time, until I realized there was a black line, too. As other reviewers have mentioned, a moving average of 12.5 hours seems very long, if you are aiming to measure subsurface water flow, which might occur on shorter timescales.*

Agreed: We changed the used colors in the figure to better focus the reader on the important parts.

Regarding the 12.5 window length that was used, we here want to reiterate that the procedure discussed in sect. 5.2.2 (and Fig. 10) is discussed in general terms and any filtering surely would need to be adapted to the processes to be observed. However, this is quite out of scope here, as we would require a detailed processes analysis for the site in order to determine which time-scales are of interest here.

On another note, Fig. 10a actually shows two flow events (A and B), and despite the large window-length they are nicely extracted from the data.

Finally, to not be misinterpreted: Surely a 12.5h window length will suppress fast signals of any type, including flow information. Therefore, if such a large window length would be required to sufficiently suppress any noise components, then we would loose these process information.

We added a few clarifying comments to Sect. 5.2.2 in which we explicitly mention that the chosen filter would not be appropriate for flow information analysis.

> Point 9: *P19 F12: There are at least six different greens. Yellow? Regardless, consider correlating the color of the measured pair, and the non-reference electrode. Alternatively,since there is no way to differentiate in the 15-16 individual curves, consider plotting all data as faint/thin black lines and only highlighting in color the few that you discuss in more detail.*

Fig. 12 is meant as an overview plot – this type of figure does not lend itself to the analysis of individual curves. We agree that by assigning individual colors to each curve the reader could be tempted to single out individual ones. Yet, plotting everything in black/gray makes is hard to see that some curves remain relatively smooth, while others show quite some dynamics.

Therefore we chose a somewhat different colormap and removed the references between colors and electrode numbers (to reduce the temptation).

> Point 10: *P20 L12: What is meant by resolved in this case? Solved or completely shown?*

We meant that the spike is not completely shown (i.e., covered by the data). We reformulated the sentence:

'Here, a positive voltage spike is observed at mark C, when only electrode 17 warmed above zero degrees, and the beginning of a negative spike at mark D, when both electrodes, 16 and 17, warmed above zero degrees.'

> Point 11: *P24 L1: Remove "huge" potential. A great tool to collect these data has been presented, but many questions persist about the use of the data.*

We agree with your statement about the many questions to be answered before SP measurements

in these environments can be actively used to infer hydrological flow information. Yet, we still are convinced that the method bears the **potential** to do this, and given the research of recent years regarding the process modeling in this field, we would classify the potential as huge (the potential has not been realized yet).

> Point 12*: In conclusion, the authors present a very detailed information about how they addressed the numerous challenges of working in such a remote environment. With some work on the organization of ideas, the manuscript could be of interest to many who are considering adding SP to ongoing research in any environment.*

Thanks, sharing our experiences in building and operating this system was our goal for the manuscript.

---

## Editor Comment (EC1) · Ciro Apollonio (Editor) · 11 Jun 2020

Very good. I'm pleased to encourage authors for a fast submission of the revised paper. You had all positive reviews, I suggest to update the paper taking into account the useful suggestions of reviewers. Go ahead. Good work.

The Associate Editor

---

## Editor Comment (EC2) · Ciro Apollonio (Editor) · 11 Jun 2020

Thank you very much for your comments and suggestions.

---

## Editor Comment (EC3) · Ciro Apollonio (Editor) · 11 Jun 2020

Thank you very much for your comments and suggestions.

---

## Editor Comment (EC4) · Ciro Apollonio (Editor) · 11 Jun 2020

Thank you very much for your voluntary and qualified comment.

---

## Author Response (AR2)

**Responses to comments for manuscript gi-2020-5**

Dear editor, dear authors,

    thank you very much for accepting the manuscript. I added one more sentence to the discussion and removed some latex code included for the review process.

Best regards

    Maximilian Weigand

[revised manuscript text omitted]